# Maternally regulated gastrulation as a source of variation contributing to cavefish forebrain evolution

**Jorge Torres-Paz[†]\*, Julien Leclercq[†], Sylvie Rétaux\***

Paris-Saclay Institute of Neuroscience, CNRS UMR9197, Université Paris-Sud, Université Paris-Saclay, Gif-sur-Yvette, France

**Abstract** Sequential developmental events, starting from the moment of fertilization, are crucial for the acquisition of animal body plan. Subtle modifications in such early events are likely to have major impacts in later morphogenesis, bringing along morphological diversification. Here, comparing the blind cave and the surface morphotypes of *Astyanax mexicanus* fish, we found heterochronies during gastrulation that produce organizer and axial mesoderm tissues with different properties (including differences in the expression of *dkk1b*) that may have contributed to cavefish brain evolution. These variations observed during gastrulation depend fully on maternal factors. The developmental evolution of retinal morphogenesis and hypothalamic patterning are among those traits that retained significant maternal influence at larval stages. Transcriptomic analysis of fertilized eggs from both morphotypes and reciprocal $F_1$ hybrids showed a strong and specific maternal signature. Our work strongly suggests that maternal effect genes and developmental heterochronies that occur during gastrulation have impacted morphological brain change during cavefish evolution.

**\*For correspondence:**
jorge.torres-paz@cnrs.fr (JT-P);
retaux@inaf.cnrs-gif.fr (SRé)

[†]These authors contributed equally to this work

**Competing interests:** The authors declare that no competing interests exist.

## Introduction

Gastrulation is a fundamental process in organism development, leading to the establishment of the embryonic germ layers (endoderm, mesoderm and ectoderm) and the basic organization of the body plan. Although in vertebrates early embryonic development has adopted highly diverse configurations, gastrulation proceeds through evolutionary conserved morphogenetic movements, including the spreading of blastoderm cells (epiboly), the internalization of mesoderm and endoderm, convergent movements towards the prospective dorsal side and extension along the antero-posterior axis (convergence and extension, respectively) (*Solnica-Krezel, 2005*). Internalization of mesendodermal cells takes place through the blastopore, which is structurally circumferential in anamniotes (fishes and amphibians) and lineal in avian and mammalian amniotes (primitive streak).

A critical step for gastrulation to proceed is the establishment of the embryonic organizer (Spemann-Mangold organizer in frogs, shield in fishes, Hensen's node in birds and node in mammals), a signaling center that is essential to instruct the formation of the body axis. In fishes and amphibians, the induction of the embryonic organizer in the prospective dorsal side occurs downstream of earlier developmental events, which are driven by maternal determinants deposited in the oocyte during maturation in the ovaries (*Kelly et al., 2000*; *Nojima et al., 2004*; *Zhang et al., 1998a*). From the organizer will emerge the axial mesoderm, a structure that spans the complete rostro-caudal extent of the embryo, with the prechordal plate anteriorly and the notochord posteriorly. The axial mesoderm is the signaling center that will induce the neural plate/tube vertically in the overlying ectoderm.

The prechordal plate is key for the patterning of the forebrain, which it affects through the regulated secretion of morphogens including sonic hedgehog (shh), Fibroblast growth factors (fgf), and

inhibitors of the Wingless-Int (Wnt) pathway, such as dickkopf1b (dkk1b) and secreted frizzled-related proteins (sFRP). Along its rostral migration, the prechordal plate is required for the sequential patterning of forebrain elements (*García-Calero et al., 2008*; *Puelles and Rubenstein, 2015*), demonstrating a temporal and spatial requirement for this migratory cell population for brain development from gastrulation onwards.

Within the central nervous system, the forebrain plays a key role in processing sensory information from the environment and controlling higher cognitive functions. During evolution and across species, different forebrain modules have experienced impressive morphological modifications according to their ecological needs, but the basic *Bauplan* to build the forebrain has been conserved. Temporal (heterochronic) and spatial (heterotopic) variation in the expression of regionalization genes and morphogens during embryogenesis have sculpted brain shapes as phylogenies have developed (*Bielen et al., 2017*; *Rétaux et al., 2013*).

An emergent model organism in which to study the impact of early embryogenesis on brain evolution at the microevolutionary scale is the characid fish *Astyanax mexicanus*. This species exists in two different eco-morphotypes that are distributed in Central and North America: a 'wild type' river-dwelling fish (surface fish) and several geographically isolated troglomorphic populations (cavefish) that live in total and permanent darkness (*Mitchell et al., 1977*; *Elliott, 2018*). Fish from the cave morphotype can be easily identified because they lack eyes and pigmentation. As a result of the absence of visual information, the cavefish has evolved mechanisms of sensory compensation, such as enhanced chemosensory and mechanosensory sensibilities (*Hinaux et al., 2016*; *Yoshizawa et al., 2010*). Sensory and other behavioral adaptations may have allowed them to increase their chances of finding food and mates in caves. Such behavioral changes are associated with morphological modifications such as larger olfactory sensory organs (*Blin et al., 2018*; *Hinaux et al., 2016*), increased number of facial mechanosensory neuromasts (*Yoshizawa et al., 2014*) and taste buds (*Varatharasan et al., 2009*), and modified serotonergic and orexinergic neurotransmission systems (*Alié et al., 2018*; *Elipot et al., 2014a*; *Jaggard et al., 2018*). Remarkably, such morphological and behavioral adaptations have a developmental origin, and have arisen mainly due to heterotopic and heterochronic differences in the expression of signaling molecules from midline organizers at the end of gastrulation, specifically at the 'neural plate' or bud stage. Subtle differences in the *shh* and *fgf8* expression domains, which are respectively larger and earlier in cavefish compared to surface fish, affect downstream processes of gene expression, morphogenetic movements during neurulation and cell differentiation, driving the developmental evolution of the cavefish nervous system (*Hinaux et al., 2016*; *Menuet et al., 2007*; *Pottin et al., 2011*; *Ren et al., 2018*; *Yamamoto et al., 2004*). As these differences in genes expressed in the midline are already manifest in embryos at the end of body axis formation, we postulated that they should stem from earlier developmental events that occur during axis formation and gastrulation.

In order to search for variations in precocious ontogenetic programs that have led to the phenotypic evolution observed in *A. mexicanus* morphotypes, we performed a systematic comparison of the gastrulation process in cave and surface embryos. We found that in the cavefish, the migration of different mesodermal cell populations is more precocious, prompting us to go further backwards in embryogenesis and to investigate maternal components. Taking advantage of the inter-fertility of the two morphotypes, we compared gastrulation, forebrain phenotypes and maternal transcriptomes in embryos obtained from reciprocal crosses between cavefish and surface fish males and females. We found that maternal factors that are present in the egg contribute greatly to the evolution of cavefish gastrulation and subsequently to forebrain developmental evolution.

## Results

### Molecular identity of the gastrula margin in *A. mexicanus*

In the zebrafish, the embryonic organizer/shield becomes morphologically evident at the prospective dorsal margin of the blastopore immediately after the epiboly has covered half of the yolk cell (50% epiboly), a stage that coincides with the initiation of the internalization of mesendodermal precursors. We studied the expression of genes involved in the establishment of the organizer in the two *A. mexicanus* morphotypes at the equivalent stage by in situ hybridization (ISH), in order to search for early differences.

First, at 50% epiboly, the inhibitor of the Wnt signaling pathway, *Dkk1b*, was expressed in a strikingly different pattern in the two morphs. In the surface fish, *dkk1b* expression was observed at the dorsal margin in two groups of cells separated by a gap in the center (*Figure 1A*), a pattern observed in the majority of the embryos (around 70%; *Figure 1C* blue). In the cavefish, a single central spot of variable extension (*Figure 1B*) was observed in most of the samples analyzed (around 70%; *Figure 1C* red). A minority of embryos of each morphotype showed an intermediate pattern corresponding to a line of positive cells without a clear interruption (not shown, *Figure 1C* green). To interpret this *dkk1b* pattern difference between the two morphs, fluorescent ISH and confocal imaging was performed. In cavefish at 50% epiboly, the *dkk1b+* cells were already internalized under

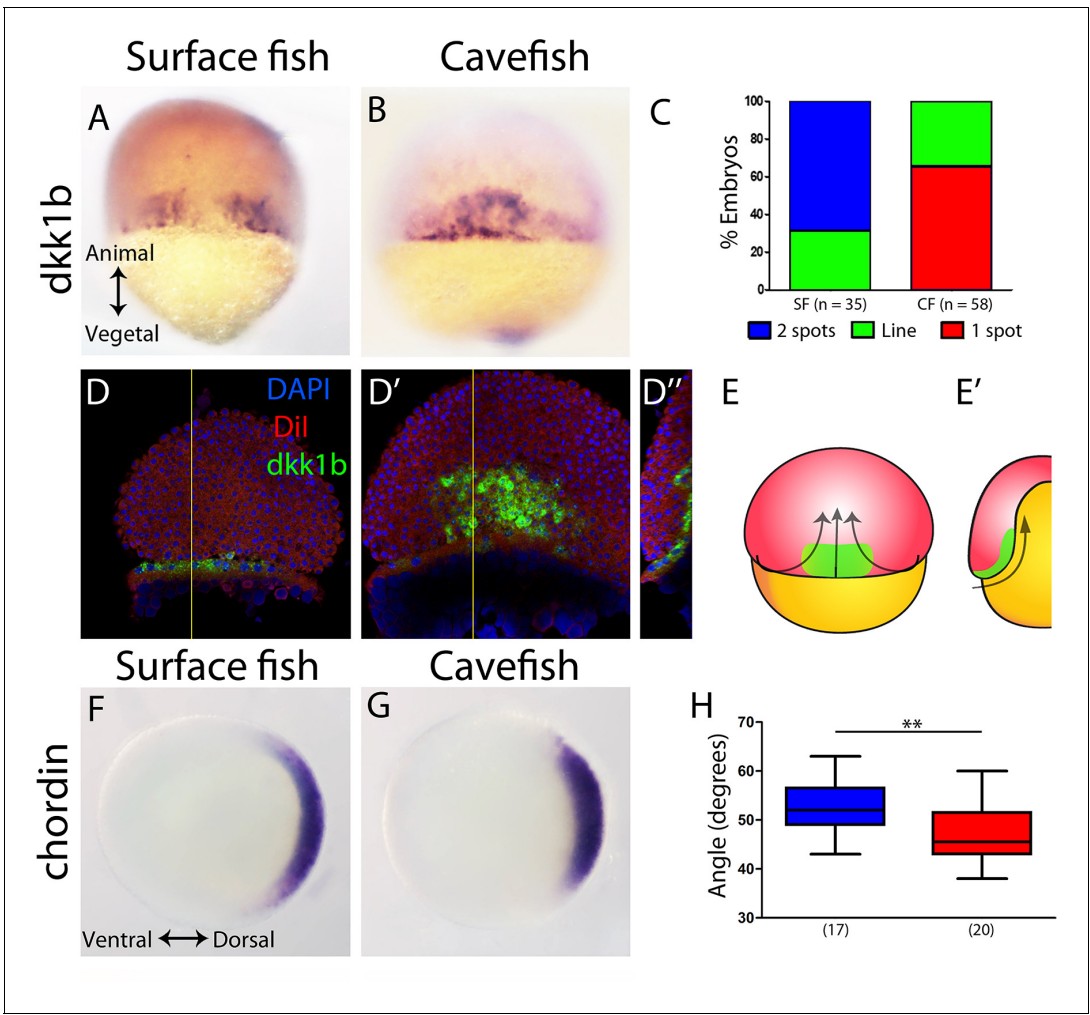

**Figure 1.** Expression of genes in the organizer at 50% epiboly in surface fish and cavefish. (A–B) Expression of *dkk1b* in surface fish (A) and cavefish (B) in dorsal view. (C) Quantification of the expression patterns observed in each morphotype. The y-axis indicates the percentage of embryos belonging to each of the categories and the number of embryos analyzed is indicated. 'Two spots' (blue) is the pattern observed in panel (A), 'one spot' (red) is the pattern observed in panel (B), and 'Line' is an intermediate profile (not shown). (D, D'') Confocal optical sections of superficial (D) and deep (D') planes and orthogonal section (D'') at the level of the yellow line in panels (D) and (D') of a cavefish embryo stained with DiI (red) and DAPI (blue) after fluorescent ISH to *dkk1b*. (E, E') Representation of the cell movements of convergence and internalization (arrows) in a dorsal view (E) and in a section (E'), with the *dkk1b+* cells represented in green. (F, G) Expression of *chordin* in surface fish (F) and cavefish (G) in animal view. (H) Quantification of the angle covered *chordin* expression pattern. Panels (A, B, D, D') are dorsal views, animal pole upwards. Panels (F, G) are animal views, dorsal to the right. The results of a Mann-Whitney test are shown in panel (H), **, p = 0.0083.

The online version of this article includes the following figure supplement(s) for figure 1:

**Figure supplement 1.** Measurement of angle in embryos stained for *chordin* at 50% of epiboly, in animal view.
**Figure supplement 2.** Expression of genes in the margin at 50% epiboly in surface fish and cavefish.

the dorsal aspect of the margin (*Figure 1D–D'' and E,E'*), revealing a precocious internalization process when compared to that in surface fish.

Chordin is a dorsalizing factor, an inhibitor of the Bmp pathway. In *A. mexicanus*, it is expressed broadly in the dorsal side (*Figure 1F,G*), similarly to the pattern in zebrafish embryos (*Langdon and Mullins, 2011*; *Miller-Bertoglio et al., 1997*). In surface fish embryos, *chordin* expression extended more ventrally than in cavefish (*Figure 1F–H*), as quantified by measuring the angle of expression in an animal view (*Figure 1—figure supplement 1*). From a dorsal view, *chordin* showed a slightly larger extension along the vegetal to animal axis, although this was not significant (not shown). This difference in *chordin* pattern extension suggested that convergence towards the dorsal pole was more advanced in cavefish.

Lefty1 is part of a feedback loop that regulates nodal signaling activity, which is involved in axial mesoderm formation and the establishment of lateral asymmetry (*Bisgrove et al., 1999*; *Meno et al., 1998*). In *A. mexicanus* embryos, *lefty1* expression was observed in the dorsal margin at 50% epiboly (*Figure 1—figure supplement 2A–B*). The ventro-dorsal extension of *lefty1* expression was similarly variable in both morphotypes at this stage (not shown) and no significant differences were observed in the vegetal-animal extension of *lefty1* expression (*Figure 1—figure supplement 2C*).

We also compared the expression of three genes that are involved in notochord development: *floating head* (*flh*), *no-tail* (*ntl*) and *brachyury* (*bra*) (*Glickman et al., 2003*; *Schulte-Merker et al., 1994*; *Talbot et al., 1995*). At 50% epiboly, the homeobox gene *flh* showed localized expression in the dorsal margin (*Figure 1—figure supplement 2D–E*), without differences in width nor in height when compared between morphotypes (*Figure 1—figure supplement 2F*). At the same stage, *ntl* and *bra* expression extended homogenously all around the margin (blastopore), hindering the identification of the prospective dorsal side (*Figure 1—figure supplement 2G,H and I,J*). No differences were observed between surface fish and cavefish.

In zebrafish, *dkk1b* is expressed in two spots in the embryonic organizer (*Hashimoto et al., 2000*) similarly to the surface fish condition (*Figure 1A*), although the gap is less pronounced in zebrafish. We reasoned that the size of the '*dkk1b* gap' may vary because of differences in dorsal convergence and the internalization of mesodermal lineages during gastrulation, relative to epiboly. The narrower domain of *chordin* expression that is observed in cavefish compared to surface fish also supported this hypothesis. To test this idea, we next analyzed the expression of axial mesodermal markers during subsequent stages of gastrulation.

## Mesoderm migration timing in *A. mexicanus* morphotypes

The EVL (enveloping layer) and YSL (yolk syncytial layer) drive epiboly movements that engulf the yolk cell (*Bruce, 2016*). Axial mesoderm precursors are mobilized from the dorsal organizer towards the rostral extreme of the embryo (animal pole), migrating inbetween the YSL and the epiblast (prospective neurectoderm). As these events are important for the induction and patterning of the neural tube, we compared in detail the process of axial mesoderm migration in *A. mexicanus* morphotypes using markers of different mesodermal populations, always taking the percentage of epiboly as a reference to stage the embryos.

The *dkk1b* patterns in the two morphs were also clearly different towards mid-gastrulation. In surface fish, the two clusters observed at 50% epiboly began to coalesce at the midline at 70% epiboly (*Figure 2A*), whereas in cavefish, *dkk1b*-expressing cells became grouped dorsally and the leading cells were more advanced towards the animal pole (arrow in *Figure 2B*). At 80% epiboly, *dkk1b*+ cells in the cavefish were close to their final position in the anterior prechordal plate at the rostral end of the embryonic axis (arrow *Figure 2E*). At the same stage, leading cells expressing *dkk1b* in the surface fish (arrow *Figure 2D*) had reached a similar distance as those in cavefish at 70% epiboly (compare values in *Figure 2F and C*). These expression profiles indicated that even though *dkk1b* expression at 50% epiboly appears very divergent in the two morphotypes, the cellular arrangements observed later on are similar, although always more advanced in the cavefish.

The same analysis was performed at 70% epiboly for the markers *chordin* (*Figure 2G–I*), *lefty1* (*Figure 2J–L*) and *ntl* (*Figure 2M–O*). These three genes showed a greater height value for their expression domain in cavefish than in surface fish embryos. This further suggested that at equivalent stages during gastrulation, anteroposterior axis formation is more advanced in cavefish.

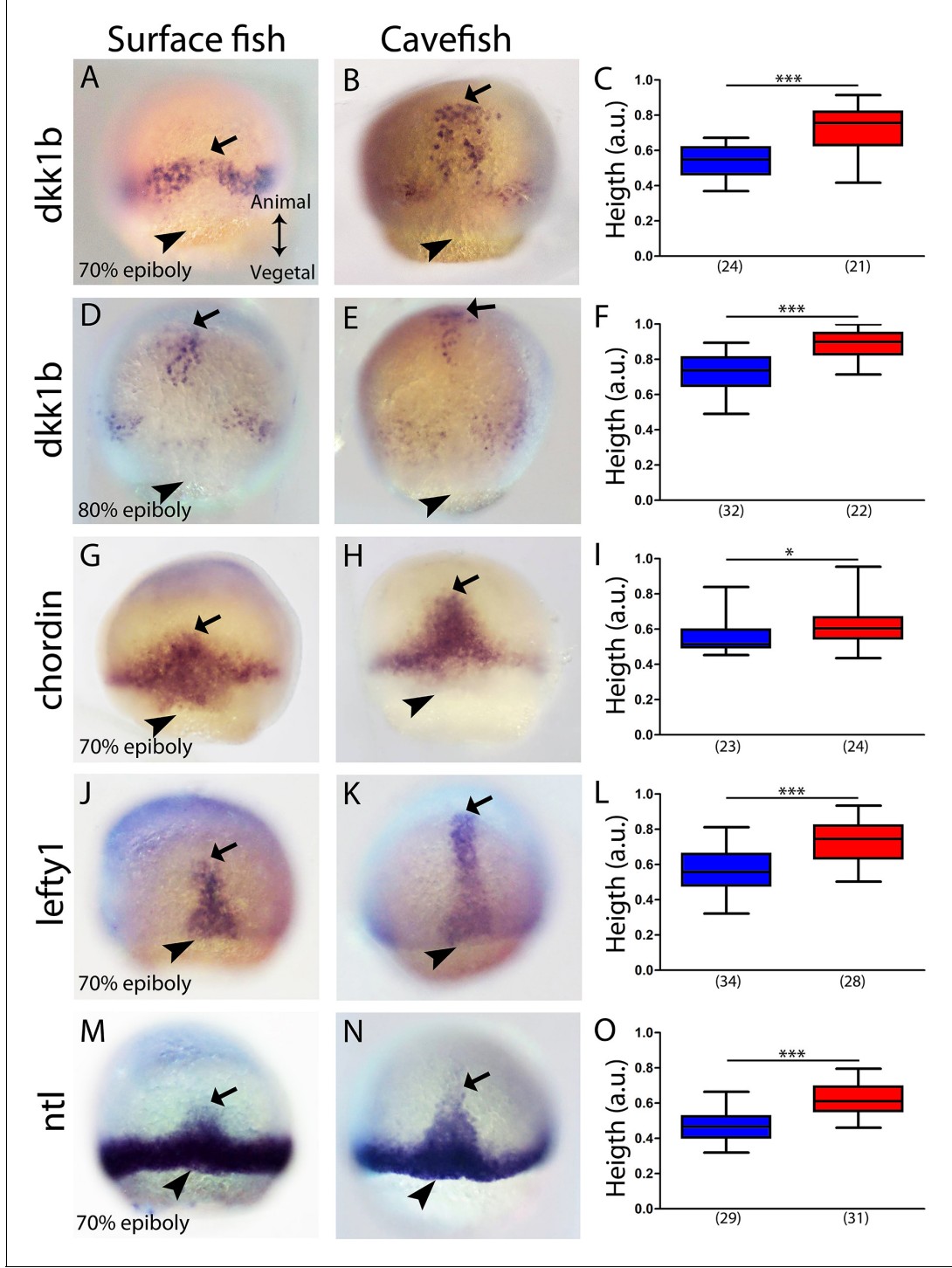

**Figure 2.** Expression of axial mesodermal genes during mid-gastrulation in surface fish and cavefish. (A, B, D, E) Expression of *dkk1b* in surface fish (A, D) and cavefish (B, E) at 70% and 80% epiboly (A, B and D, E, respectively). (C, F) Quantification of height (see *Figure 2—figure supplement 1*) in *dkk1b*-labeled embryos at 70% and 80% epiboly (C and F, respectively). (G, H) Expression of *chordin* in surface fish (G) and cavefish (H) at 70% epiboly. (I) Quantification of height in *chordin*-labeled embryos at 70% epiboly. (J, K) Expression of *lefty1* in surface fish (J) and cavefish (K) at 70% epiboly. (L) Quantification of height in *lefty1*-labeled embryos at 70% epiboly. (M, N) Expression of *ntl* in surface fish (M) and cavefish (N) at 70% epiboly. (O) Quantification of height in *ntl*-labeled embryos at 70% epiboly. Embryos in dorsal views, animal pole upwards. Mann-Whitney test were performed. ***, p = <0.0001, *, p = 0.0167.

The online version of this article includes the following figure supplement(s) for figure 2:

**Figure supplement 1.** Measurement of the height,.

Next, we wondered whether the observed phenotype for the cavefish axial mesoderm also extends to the neighboring paraxial mesoderm, that is the mesodermal tissue located laterally that will give rise to the somites (presomitic mesoderm). We analyzed the expression of *myoD* and *mesogenin 1* (*msgn1*), two genes coding for bHLH transcription factors that are required for early specification of myogenic tissue (*Weinberg et al., 1996*; *Yabe and Takada, 2012*). In *A. mexicanus*, at mid-gastrulation *myoD* was expressed in two domains, each triangular in shape and positioned on either side of the dorsal axial mesoderm, which corresponds to the central gap that is without expression (*Figure 3A–D*). The height value of the expression domain was higher in cavefish embryos, at both at 70% and 80% epiboly, than in surface fish embryos (*Figure 3E*), whereas the central/dorsal non-expressing zone was wider in the surface fish at both stages (*Figure 3F*). On the other hand, at the same stages, *msgn1* extended as a ring all around the margin except on its dorsal aspect, leaving a central gap (*Figure 3G–J*). For *msgn1*, no significant differences were found in the height values at 70% and 80% epiboly (*Figure 3K*), but similarly to what was observed for *myoD*, the dorsal non-expressing zone was reduced in cavefish embryos at 80% epiboly (*Figure 3L*). In order to understand the inter-morph differences observed using these two paraxial mesoderm markers, we performed double ISH. As in single ISH, *msgn1* expression extended further ventrally than *myoD* (*Figure 3M, O*; compare to insets in *Figure 3A, B, G and H*). Differences also existed in the vegetal to animal axis, where the larger extension encompassed by *myoD* was clear in both morphs (*Figure 3M, O*). These results suggested that the differences observed in our measurements of paraxial mesoderm extension were mainly due to the expression of *myoD* (but not *msgn1*) in the cell population, which is more advanced towards the animal end of the embryo (*Figure 3N, P*). In addition, if the size of the central zone where expression of the two paraxial markers is interrupted is taken as readout of dorsal convergence, these data also suggest an earlier convergence and extension in cavefish than in surface fish (at a given stage of epiboly).

## *A. mexicanus* morphotypes exhibit notable differences in axial mesoderm structure

The antero-posterior embryonic axis in *A. mexicanus* is formed after epiboly has been completed, at the bud-stage (10 hours post-fertilization [hpf]). The prechordal plate and notochord are the anterior and posterior segments of the axial mesoderm, respectively, and both are important for the induction and patterning of neural fates. To compare the organization of the axial mesoderm in cave and surface embryos, we analyzed the expression of markers described in the previous sections to identify specific segments once the antero-posterior axis has been formed. Using triple fluorescent in situ hybridization, three non-overlapping molecular subdomains were recognized: the anterior prechordal plate or polster labeled by *dkk1b*, the posterior prechordal plate defined by *shh* expression (wider in cavefish as previously described; *Pottin et al., 2011*; *Yamamoto et al., 2004*) and the notochord more posteriorly, labeled by *ntl* (*Figure 4A,B*). In addition, *lefty1* expression covered both the anterior and posterior subdomains of the prechordal plate (*Figure 4C–F*). In the posterior prechordal plate, *lefty1* and *shh* showed overlapping patterns in both morphotypes (*Figure 4C,D*), whereas *dkk1b* and *lefty1* showed only minimal co-expression anteriorly (*Figure 4E,F*), similar to the pattern observed at earlier stages (*Figure 4—figure supplement 1*). Moreover, the distribution of polster *dkk1b*-expressing cells was strikingly different between the two morphs. In surface fish, they were tightly compacted (*Figure 4A*), whereas in cavefish, they were loosely organized (*Figure 4B*). The numbers of *dkk1b*-expressing cells, analyzed in confocal sections, were similar in cavefish and surface fish (*Figure 4G*). The distribution of the *dkk1b* cells in the antero-posterior axis, measured by the distance between the first and the last cells (Length A-P), was identical (*Figure 4H*). However, the *dkk1b*-positive cells covered a larger extension in the lateral axis (Length lateral) in cavefish embryos (*Figure 4I*), indicating that these cells are arranged at a lower density than in surface fish. A similar pattern was observed for the anterior domain of *lefty1* expression (compare *Figure 4C, E and Figure 4D, F*). Thus, both the anterior/polster (*dkk1b+*) and the posterior part (*shh+*) of the prechordal plate are laterally expanded in cavefish.

Next, other differences in the size or position of axial mesoderm segments at the bud stage were explored. The distance from the anterior-most polster cell expressing *dkk1b* to the leading notochord cell expressing *ntl* was identical in the two morphs (*Figure 4—figure supplement 2A–C*). Polster cells expressing *dkk1b* laid just beneath the cells of the anterior neural plate border (*dlx3b+*) in both morphotypes (*Figure 4—figure supplement 2D,E*). The extension of the notochord was also

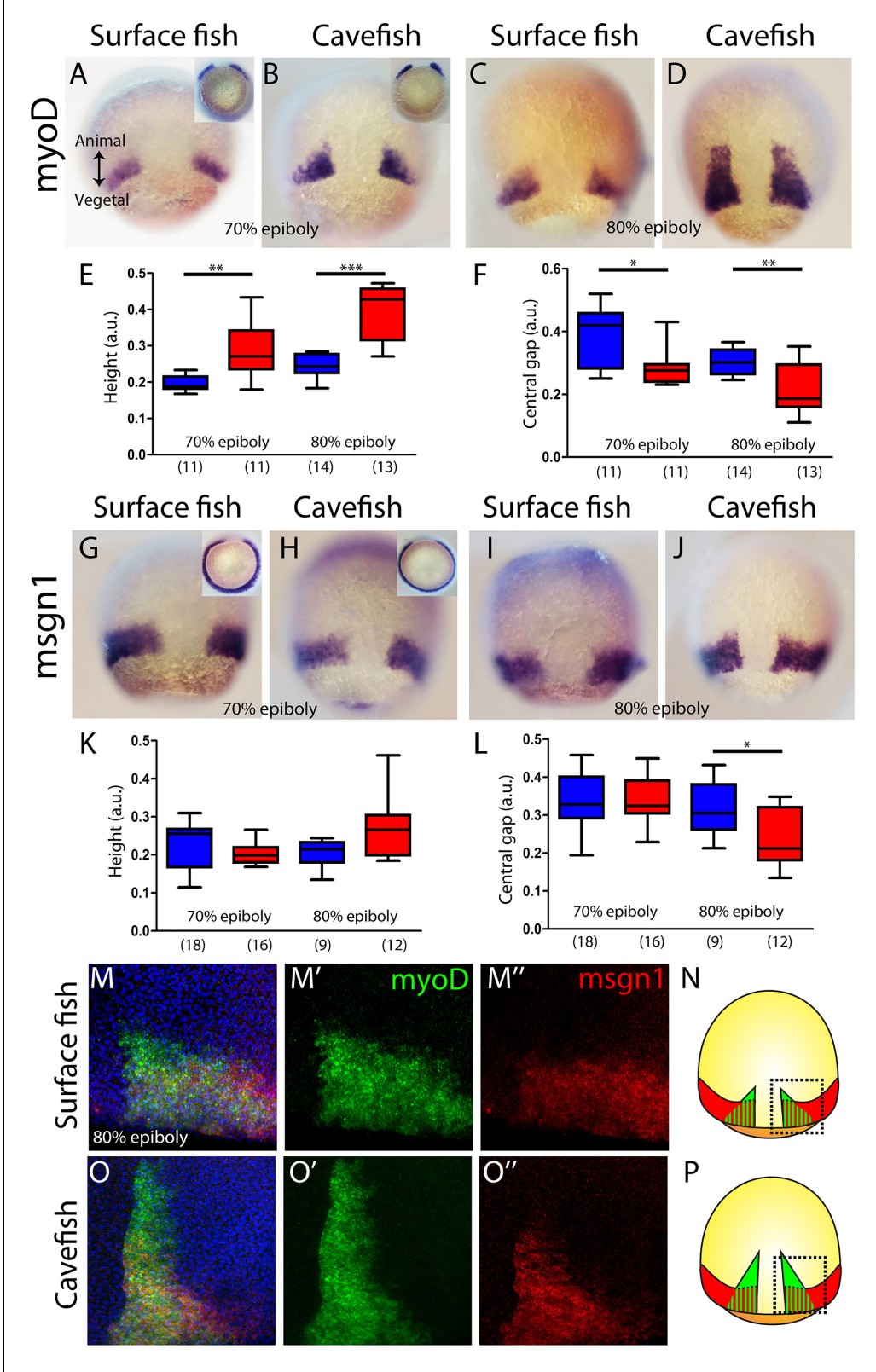

**Figure 3.** Internalization of paraxial mesoderm in surface fish and cavefish. (**A–D**) Expression of *myoD* in surface fish (**A, C**) and cavefish (**B, D**) at 70% and 80% epiboly (A, B and C, D, respectively). Insets in panel (**A**) and (**B**) show the corresponding embryos in a vegetal view. (**E**) Quantification of height in *myoD*-labeled embryos at 70% and 80% epiboly (left and right, respectively). (**F**) Quantification of the central non-expressing zone in *myoD*-labeled embryos at 70% and 80% epiboly (left and right, respectively). (**G–J**) Expression of *msgn1* in surface fish (**G, I**) and cavefish (**H, J**) at 70% and 80%

*Figure 3 continued on next page*

*Figure 3 continued*

epiboly (G, H and I, J, respectively). Insets in panels (G) and (H) show the corresponding embryos in a vegetal view. (K) Quantification of height in *msgn1*-labeled embryos at 70% and 80% epiboly (left and right, respectively). (F) Quantification of the central non-expressing zone in *msgn1*-labeled embryos at 70% and 80% epiboly (left and right, respectively). (M–M'' and O–O'') Confocal projection (20–30 μm) showing the expression of *myoD* (green) and *msgn1* (red) in double-stained surface fish and cavefish embryos (M–M'' and O–O'', respectively) at 80% epiboly. DAPI was used as a counterstain (blue nuclei). (N, P) representations of surface fish (N) and cavefish (P) embryos, indicating in black dashed lines the regions of interest showed in panels (M) and (O). Mann-Whitney tests were performed. **, p = 0.0025 (E, left), ***, p = <0.0001 (E, right), *, p = 0.0181 (F, left), **, p = 0.0094 (F, right), *, p = 0.0209 (L, right). Embryos in dorsal views, animal pole on top; insets in vegetal view, dorsal on top.

The online version of this article includes the following figure supplement(s) for figure 3:

**Figure supplement 1.** Measurement of the width.

measured. At bud stage, *ntl* and *bra* expression labeled the notochord in its whole extension (*Figure 4J,K* and not shown). On the other hand, *flh* was expressed in the posterior end and in a small cluster of the rostral notochord (*Figure 4L,M*) (as well as in two bilateral patches in the neural plate that probably correspond to the prospective pineal gland in the diencephalon). For the three notochord markers, the distance from the rostral expression boundary to the tail bud (normalized by the size of the embryo) was larger in cavefish than in surface fish (*Figure 4N–P*). In line with our observations of axial and paraxial mesoderm markers during mid-gastrulation (*Figures 2–3*), these results suggest a precocious convergence and extension in cavefish when compared to surface fish.

## Testing the effects of heterochrony in gastrulation and gene-expression dynamics on brain development

In zebrafish embryos, *dkk1b* expression in the prechordal plate becomes downregulated from early somitogenesis (*Hashimoto et al., 2000*). Our observations of heterochronic gastrulation events prompted us to search for potential differences in the disappearance of *dkk1b* expression later on. In surface fish, *dkk1b* was still expressed in all embryos at the 6- and 8-somite stage (13/13 [not shown] and 17/17 [*Figures 5A* and *6E*, right]). By contrast, in cavefish, *dkk1b* expression was observed only in 46% of the embryos at the 6-somite stage (6/13, always with low signal level; not shown) and it was absent in 64% of embryos at the 8-somite stage (21/33, *Figures 5B* and *6E* right).

Given the major spatio-temporal differences in *dkk1b* expression pattern observed from the onset of gastrulation to the end of neurulation between cave and surface embryos, we also examined the expression levels of *dkk1b* by qPCR. At 50% epiboly, *dkk1b* transcript levels were similar in the two morphs (0.95 fold, NS), but at bud stage, *dkk1b* levels were almost four times lower in cavefish than in surface fish embryos (0.27 fold).

As *dkk1b* is a strong inhibitor of Wnt signaling, with conserved functions in the regulation of brain development (*Hashimoto et al., 2000*; *Lewis et al., 2008*), the observed differences in the cellular arrangement, expression levels and timing of downregulation of *dkk1b* in the two *Astyanax* morphotypes may have downstream consequences in forebrain morphogenesis. This hypothesis was partly tested by treating surface fish embryos with LiCl (*Figure 5C*), a Wnt-βcat pathway activator, to mimic the cavefish situation in which the Wnt antagonist *dkk1b* is expressed at lower levels. In line with results reported in zebrafish (*Shinya et al., 2000*), LiCl treatments (0.1M and 0.2M, from 8 to 13 hpf) in surface fish produced a decrease of the size of the optic vesicle at 13 hpf (not shown) and a reduction of the size of the retina and lens at 24 hpf (*Figure 5D–F,H,I*), which are hallmarks of cavefish embryonic eye morphology (*Yamamoto et al., 2004*); compare to *Figure 5G*. In addition, manipulation of the levels of Wnt-βcat signaling in surface fish produced a misshaped retina with a wider optic stalk (*Figure 5F*). This was observed in 41% and 50% of the examined eyes of embryos treated with LiCl 0.1M and 0.2M, respectively (*Figure 5J*). A similar phenotype was seen in 23% of cavefish embryos at the same stage (*Figure 5G,J*). The interpretation of this morphological coloboma-like phenotype was confirmed molecularly, as the expression domain of the optic stalk marker *pax2a* was significantly wider at 36 hpf in surface fish embryos exposed to LiCL and in cavefish embryos than in untreated surface fish embryos (*Figure 5K–N,O*).

Together these data strongly suggest that modified levels of Wnt signaling during early embryogenesis might contribute to the developmental evolution of cavefish eye defects.

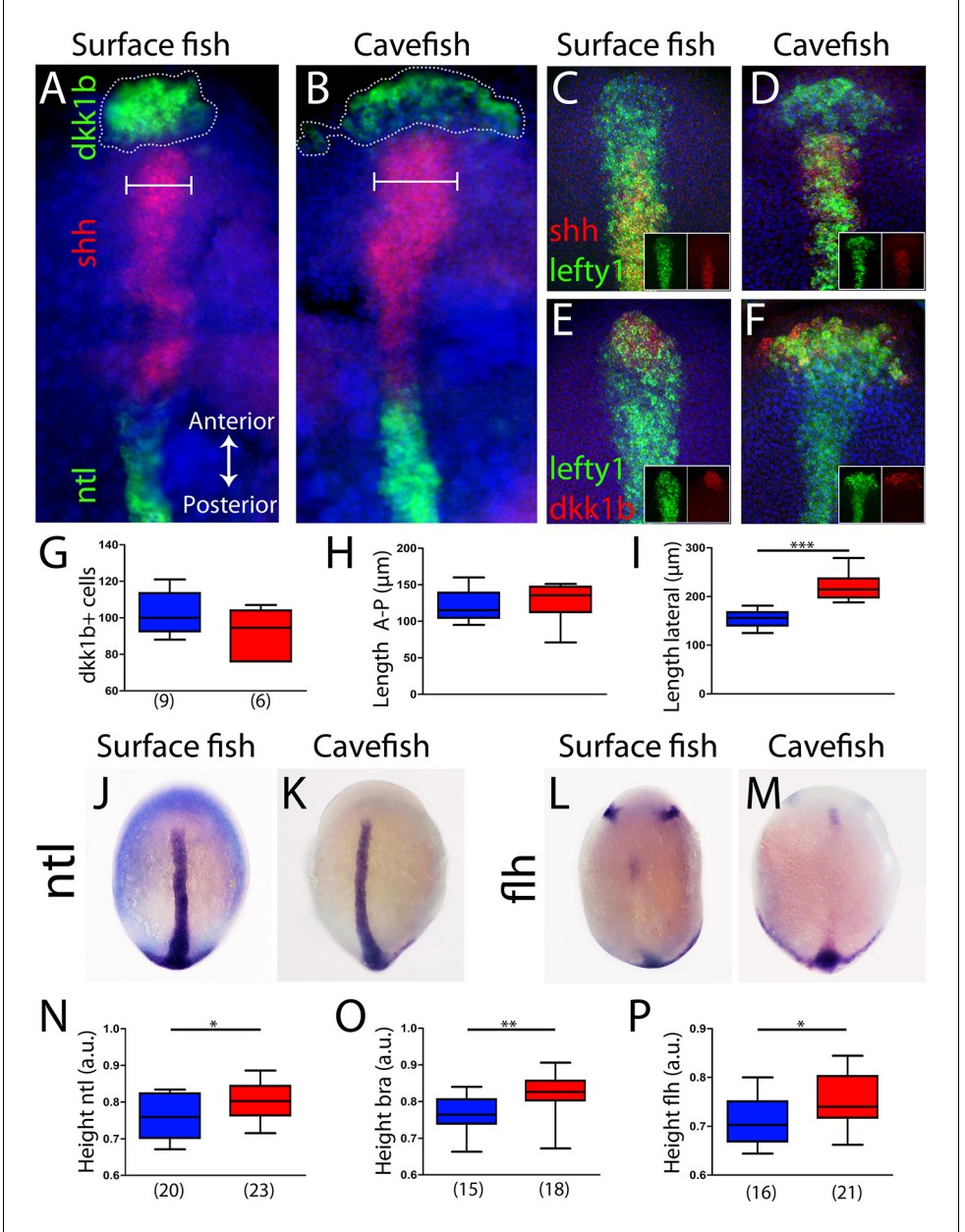

**Figure 4.** Axial mesoderm organization in surface fish and cavefish. (A, B) Triple ISH to *dkk1b* (green, rostral), *shh* (red, central) and *ntl* (green, posterior) in surface fish (A) and cavefish (B). (C, D) Confocal projection (20–30 μm) showing the expression of *shh* (red) and *lefty1* (green) in surface fish (C) and cavefish (D) embryos. Insets show the individual channels. (E, F) Confocal projection (20–30 μm) showing the expression of *dkk1b* (red) and *lefty1* (green) in surface fish (E) and cavefish (F) embryos. Insets show the split channels. (G) Quantification of the number of cells expressing *dkk1b*. (H) Quantification of the distance between the *dkk1b*-expressing cells located in the extremes of the antero-posterior axis. (I) Quantification of the distance between the *dkk1b*-expressing cells in lateral extremes. (J, K) Expression of *ntl* in surface fish (J) and cavefish (K). (L, M) Expression of *flh* in surface fish (L) and cavefish (M). (N) Quantification of height in *ntl*-labeled embryos. (O) Quantification of height in *bra*-labeled embryos. (P) Quantification of height in *flh*-labeled embryos. All embryos at tail-bud stage, in dorsal view, anterior upwards. Pictures in panels (A–F) are flat mounted embryos, whereas pictures in panels (J–M) are whole-mount embryos. Mann-Whitney tests were performed. ***, p = <0.0001 (I); *, p = 0.0396 (N); **, p = 0.0012 (O); and *, p = 0.0142 (P).

The online version of this article includes the following figure supplement(s) for figure 4:

**Figure supplement 1.** Expression of *dkk1b* and *lefty1* during mid-gastrulation.

**Figure supplement 2.** Position of the prechordal plate relative to notochord and anterior neural border.

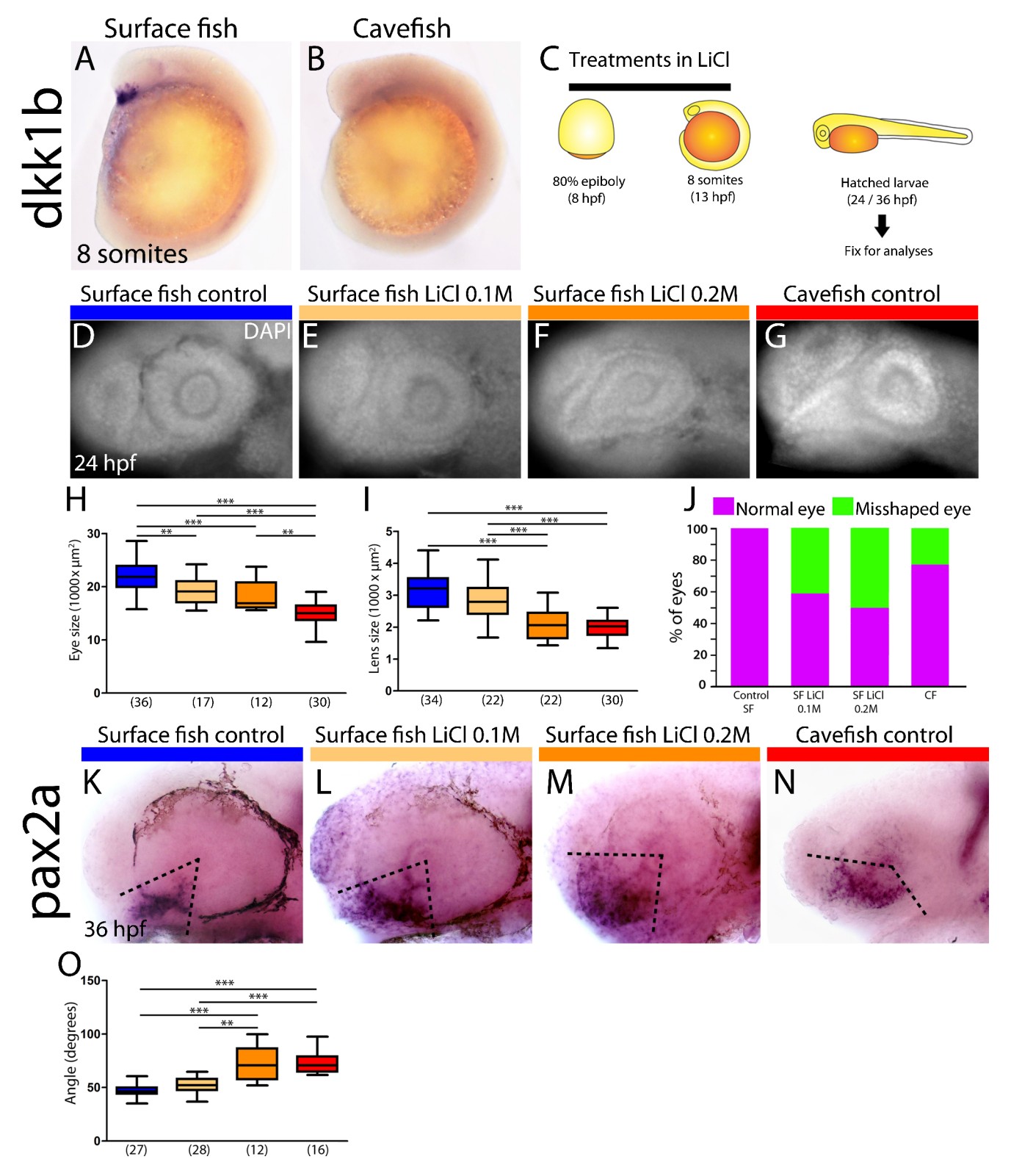

**Figure 5.** Differential off-set of *dkk1b* expression may be relevant for the optic phenotype in cavefish. (A, B) Expression of *dkk1b* at the 8-somite stage in surface fish (A) and cavefish (B). (C) Experimental design for LiCl treatments. Dechorionated surface fish embryos were treated in LiCl solutions from the end of gastrulation (8 hpf, left) until mid-somitogenesis (13 hpf, center) and then fixed for analyses at larvae stages (24 or 36 hpf, right). (D–J) Effect of LiCl treatments analyzed at 24 hpf. Surface fish untreated (D), treated with 0.1M and 0.2M LiCl (E and F, respectively) and cavefish untreated (G),

*Figure 5 continued on next page*

*Figure 5 continued*

stained with DAPI at 24 hpf. Quantification of the eye size (H) lens size (I) and percentage of embryos with misshaped developing eye (J). (K–O) Effect of LiCl treatments analyzed at 36 hpf. Expression of *pax2a* at 36 hpf in the optic stalk/optic fissure of surface fish that were untreated (K) or treated with0.1M and 0.2M LiCl (L and M, respectively) and of untreated cavefish (N). (O) Quantification of the measured angle (indicated in K-N as black dashed lines). Kruskal-Wallis tests with Dunn's post-test, were performed. **, p = <0.01; ***, p = <0.001.

The online version of this article includes the following figure supplement(s) for figure 5:

**Figure supplement 1.** Measurement of angle in embryos stained for *pax2a* at 36 hpf, in lateral view.

## Maternal determinants influence early developmental differences in *A. mexicanus* morphotypes

The earliest developmental events—including the first cell divisions, breaking of symmetries and induction of the embryonic organizer—rely exclusively on maternal factors that are deposited in the oocyte before fertilization. The findings described above, showing earlier convergence, extension and internalization of mesodermal cell populations in the cave morphs, together with differences in spatio-temporal gene regulation in tissues derived from the organizer, prompted the examination of precocious embryogenesis and the investigation of maternal components. The inter-fertility between *A. mexicanus* morphotypes offers a powerful system in which to study the potential contribution of these maternally produced factors to phenotypic evolution (*Ma et al., 2018*). We compared gastrulation progression in $F_1$ hybrid embryos obtained from the fertilization of surface fish eggs with cavefish sperm (HybSF) and from cavefish eggs with surface fish sperm (HybCF) (*Figure 6A*). In principle, phenotypic correspondence to the maternal morphotype indicates a strong maternal effect. Results obtained in $F_1$ hybrids were compared to those obtained from wild-type morphs in previous sections.

First, the expression patterns of *dkk1b* during development were compared. At 50% epiboly, the percentages each of the phenotypic categories (described in *Figure 1A–C*) present in hybrid embryos were strikingly similar to those of their maternal morphotypes, with the majority of HybSF presenting two spots of *dkk1b* expression like surface fish embryos, whereas most of HybCF embryos showed only one continuous expression domain (*Figure 6B*). At 70% epiboly, the results followed the same trend. In HybSF embryos, the two domains of *dkk1b*-expressing cells begin to join dorsally, with little advancement towards the animal pole, as in surface embryos (*Figure 6C*). By contrast, HybCF were more like cavefish embryos, with cells grouped dorsally close to the animal end (*Figure 6C*). Analyses of the distance reached by the leading cell showed significant differences between the two reciprocal hybrids types, which were identical to their maternal morphs (*Figure 6C*, right). The expression of *lefty1* and *ntl* at 70% epiboly was also examined in $F_1$ hybrids (*Figure 6— figure supplement 1A and B*, respectively). The advancement of axial mesoderm populations labeled by the two markers was significantly increased in HybCF compared to HybSF, with height values akin to those of their respective maternal morphs (*Figure 6—figure supplement 1A and B*, right). These results indicate that spatio-temporal differences observed during gastrulation between cavefish and surface fish fully depend on maternal contribution.

In *A. mexicanus*, the prechordal plate at the end of gastrulation showed marked morphotype-specific differences in cell organization. We evaluated the impact of maternal determinants on these differences (*Figure 6D*). We found a broader distribution of *dkk1b*-expressing cells in the HybCF (*Figure 6D*, center bottom) than in the HybSF (*Figure 6D*, left bottom). The patterns observed in the $F_1$ hybrids were identical those patterns in their maternal morphs (*Figure 6D*, right), highlighting the effect of the oocyte composition up to the end of gastrulation, well after the activation of the zygotic genome.

Next, we tested the maternal contribution to the disappearance of *dkk1b* expression during mid-somitogenesis. At the 8-somite stage, the segregation of phenotypes in reciprocal $F_1$ hybrids was not as clear as in the parental morphs (*Figure 6E*). For this reason, we decided to classify the expression patterns of *dkk1b* into four categories: I, widely expressed in the prechordal plate (*Figure 6E*, top left; blue); II, clear expression in 3–5 cells (*Figure 6E*, top right; green); III, clear expression in 1–2 cells (*Figure 6E*, bottom left; yellow); and IV, absence of expression (*Figure 6E*, bottom right; red). In hybrids, we found similar percentages of intermediate categories II and III (63–64% in both cases). In HybCF, however, there was an important proportion of category IV embryos (no

expression, 16%), closer to the pattern seen in cavefish, whereas none of the HybSF fell into this category, as was also the case for surface fish embryos. We concluded that at the 8-somite stage the downregulation of *dkk1b* expression is still under the influence of maternal factors, although this influence is weaker than at earlier stages.

## Maternal determinants influence late phenotypes in *A. mexicanus* morphotypes

Finally, through analyses in reciprocal $F_1$ hybrids between 15 hpf and 72 hpf, we sought to test whether maternally controlled differences in gastrulation translate into the eye and forebrain phenotypes previously described in cavefish at later embryonic and larval stages. In order to help the visualization and interpretation of the $F_1$ hybrid data, simplified plots were generated (*Figures 7* and *8*, to the right of each graph) with the mean values for cavefish and surface fish in the extremes (red and blue dots, respectively), an average black dot representing the expected value for the phenotype if there is no effect of any kind (maternal, paternal or allelic dominance), and the HybSF and HybCF values (light blue and pink, respectively). If experimental values are closer to the maternal morphotype and if reciprocal hybrids show significantly different values, this would suggest that the phenotype is under maternal regulation. Other possibilities, such as a mix of maternal and zygotic influence, or recessive or dominant effects in heterozygotes, can also be interpreted.

The most striking cavefish phenotype concerns the eye. Both the retina and the lens, two structures of different embryonic origins, are affected. The placode-derived lens is small and undergoes apoptosis (the latter being independent of the former; *Hinaux et al., 2017*) while the neural plate-derived retina is small and displays a coloboma-like morphogenetic defect (*Devos et al., 2019*; *Hinaux et al., 2016*; *Pottin et al., 2011*; *Yamamoto and Jeffery, 2000*; *Yamamoto et al., 2004*).

The difference in lens size between surface fish and cavefish was significant, as expected, and increased as development proceeds (in cavefish: −42% at 24 hpf; −66% at 2 dpf; and −84% at 3 dpf), following the regressive process in one morph and the growth process in the other (*Figure 7A–D*). However, at all three stages, lens size was identical in reciprocal $F_1$ hybrids, showing intermediate sizes between parental values (*Figure 7A–D*). This suggests that lens size is not under maternal control. Of note, the size of the olfactory placodes, which is inversely correlated to the size of lens placodes in the two *Astyanax* morphs (*Hinaux et al., 2016*), also shows no evidence of maternal influence in reciprocal $F_1$ hybrids at 24 hpf (*Figure 7—figure supplement 1*), suggesting that the developmental trade-off between the two placodal sensory derivatives is not maternally controlled.

To assess a potential maternal genetic influence on the visual degenerative process, we counted the number of Caspase3-positive apoptotic cells in the lenses of the four types of larvae (*Figure 7A, E and F*). HybSF and HybCF were indistinguishable, both showing ~2.5 apoptotic cells per $10^4$ $\mu m^3$ at 2 dpf and ~1 apoptotic cell per $10^3$ $\mu m^3$ at 3 dpf (*Figure 7E and F*). These values were closer to those of surface fish than to those of cavefish values, indicating a dominant effect of surface alleles in heterozygotes (*Figure 7E and F*, right). The exact same pattern was observed for Caspase3-positive cells in the 3 dpf retina (*Figure 7A and G*). These results show that the eye degenerative process that is associated with and triggered by lens apoptosis is not maternally controlled in cavefish. As cavefish lens apoptosis has been shown to be induced by ventral midline *Shh* overexpression (*Yamamoto et al., 2004*), we sought to check *Shh* expression phenotypes in surface fish, cavefish and their reciprocal $F_1$ hybrids at the end of gastrulation. Strikingly, and contrarily to the *Dkk1b* phenotype shown above, the width of the *Shh* expression domain in the prechordal plate at 10 hpf was identical in HybSF and HybCF (*Figure 7H and I*), suggesting a lack of maternal contribution to the control of *Shh* expression, in line with the lack of maternal control on the downstream degenerative phenotype.

We obtained markedly different results regarding retina morphogenesis. The coloboma-like defect, which includes a disorganized and wide expression of *Pax2a* at the optic stalk and ventral fissure in cavefish (*Devos et al., 2019*), was examined in reciprocal $F_1$ hybrids at 48 hpf (*Figure 7J and K*). As regards the size of the optic fissure, HybSF and HybCF were significantly different, indicating a maternal genetic effect on this phenotype, and both were closer to the surface fish values (*Figure 7K*), suggesting a partial dominance of the surface fish zygotic program. Together with the above results showing that *Dkk1b* expression in the prechordal plate is maternally controlled (*Figure 6*), and that manipulation of the Wnt signaling pathway in surface fish mimics the cavefish

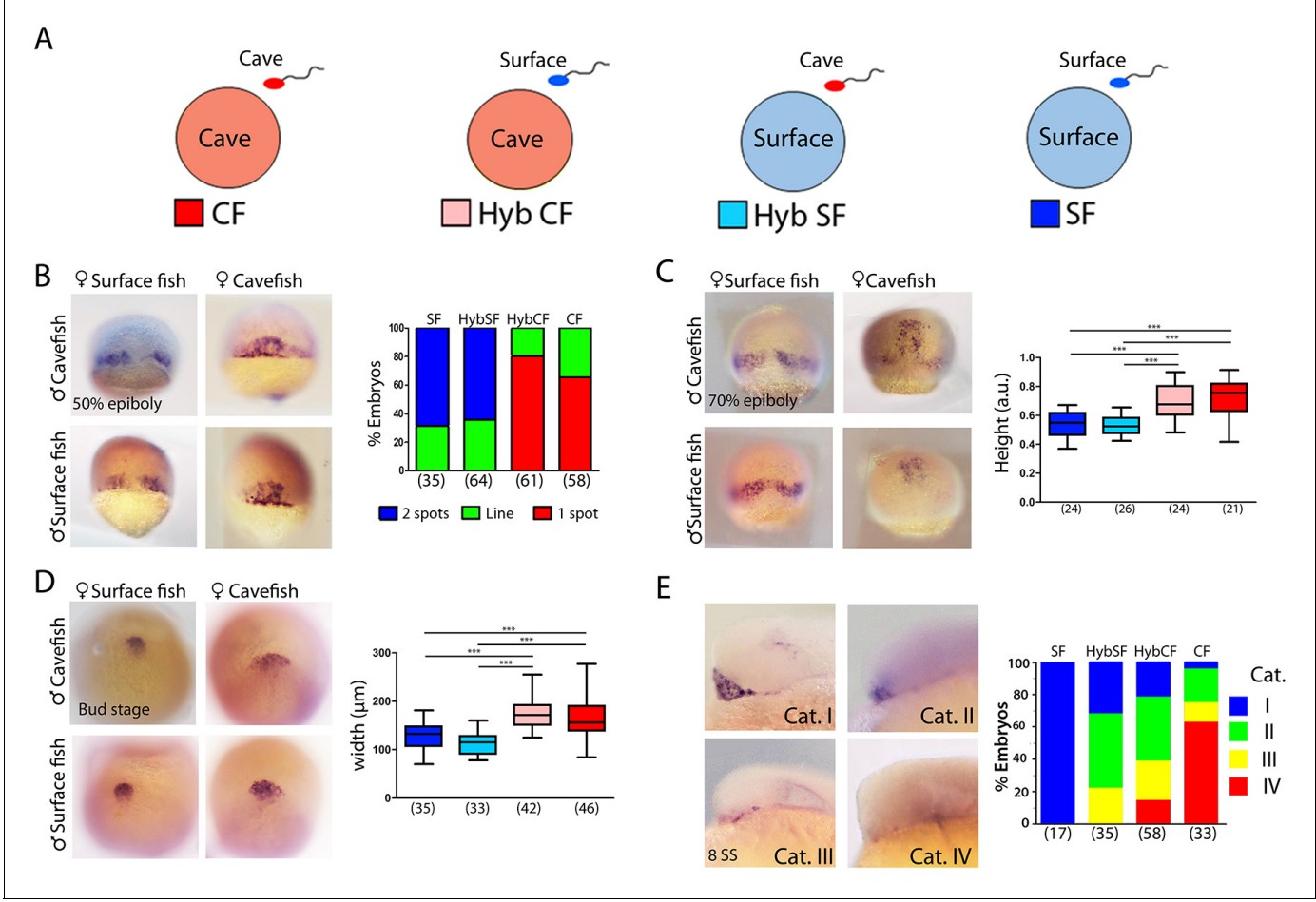

**Figure 6.** Maternal effect on early development. (A) Schematic representation of the fertilizations performed for the analyses of maternal effect in $F_1$ hybrids. Oocytes from either morph (cave in pink and surface in light blue) were fertilized with sperm from cavefish (red) or surface fish (blue). For simplicity, $F_1$ hybrids were named HybCF (oocyte from cavefish, pink) and HybSF (oocyte from surface fish, light blue), based on their maternal contribution. (B–E) Expression of *dkk1b* at 50% of epiboly (B), 70% of epiboly (C), bud stage (D) and 8-somite stage (E). (B, right) Quantification of the expression pattern of *dkk1b* at 50% epiboly, classified into three categories: '2 spots','1 spot' (red), and 'Line', which is an intermediate profile (not shown in micrographs). The y-axis indicates the percentage of the total embryos belonging to each of the three categories and the numbers of embryos examined are indicated. (C, right) Quantification of height in *dkk1b*-labeled embryos at 70% epiboly . (D, right) Quantification of width of the polster based on *dkk1b* expression. (E, right) Quantification of the pattern of *dkk1b* at the 8-somite stage, with embryos classified according to the number of positive cells. Category I (blue, surface fish), more than 5 cells; category II (green, HybSF), between 3 and 5 cells; category III (yellow, HybCF); and category IV, no positive cells (red, cavefish). All embryos were imaged in whole mount. Embryos in panels (B) and (C) in dorsal view with the animal pole upwards; embryos in panel (D) are in dorsal view with the anterior upwards; and embryos in panel (E) are in lateral view with the anterior to the left. Kruskal-Wallis tests with Dunn's post-test were performed in all cases. ***, p = <0.001.

The online version of this article includes the following figure supplement(s) for figure 6:

**Figure supplement 1.** Maternal effect during mid-gastrulation.

coloboma (*Figure 5*), these data show how the maternal control of gastrulation has long-lasting consequences for morphological evolution.

As the retina, which derives from the eyefield in the anterior neural plate, is under maternal control for its patterning, we further examined other forebrain phenotypes including those in the hypothalamus. Indeed, inter-morph variations in the expression domains of the LIM-homeodomain transcription factors *Lhx9* and *Lhx7* drive changes in Hypocretin and NPY neuropeptidergic patterning in the hypothalamus, respectively (*Alié et al., 2018*). We therefore compared the expression domains of *Lhx9* (the size of the hypothalamic domain at 15 hpf; brackets in *Figure 8A*, left) and *Lhx7* (number of positive cells at 24 hpf in the hypothalamic acroterminal domain; dotted circles in *Figure 8B*, left), as well as the numbers of their respective neuropeptidergic Hypocretin and NPY

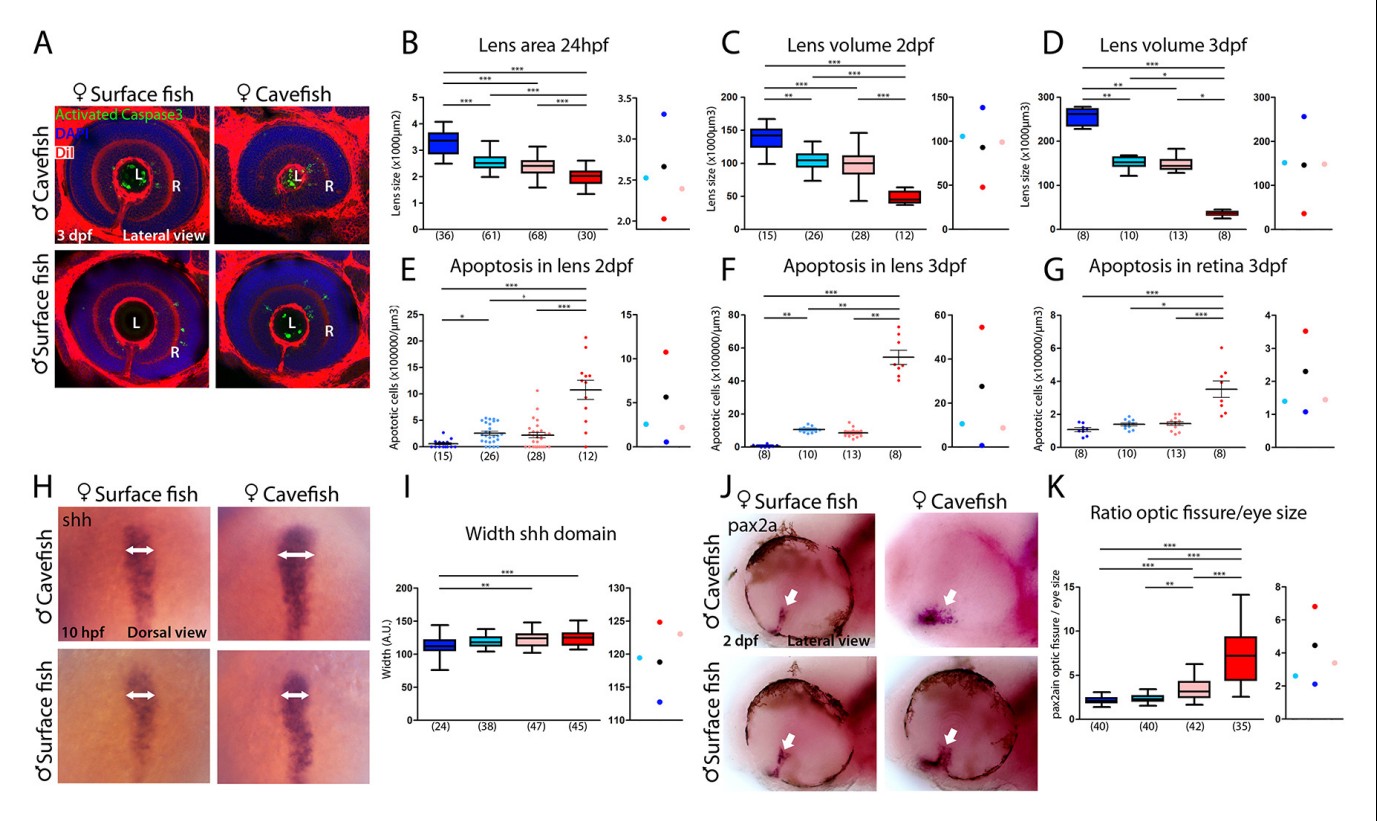

**Figure 7.** Maternal effect on eye development. (A) Immunostaining of activated caspase three in the developing eye at 3 days post-fertilization (dpf). (B–D) Quantification of the size of the lens at 24 hpf (B, area), 2 dpf (C, volume) and 3 dpf (D, volume). (E–G) Quantification of the number of apoptotic cells per unit of volume in the lens at 2 dpf (E) and 3 dpf (F) and in the retina at 3 dpf (G). (H) Expression of *Shh* at 10 hpf. (I) Quantification of the width of the *Shh* expression domain (arrows in panel H). (J) Expression of *Pax2a* in the optic fissure (arrow) at 2 dpf. (K) Quantification of the size (area) of the *Pax2a* expression domain in the optic fissure normalized by the eye size (area). Embryos are arranged in panels (A, H and J) as follows: HybSF (top left), cavefish (top right), surface fish (bottom left) and HybCF (bottom right). In the quantifications, HybSF, cavefish, surface fish and HybCF are colored light blue, red, blue and pink, respectively. A plot of means is shown to the right of each graph. Images in panels (A) and (J) correspond to lateral views of the eye, anterior to the left. Images in panel (H) are whole-mounted embryos in dorsal views, anterior to the top. Kruskal-Wallis tests with Dunn's post-test were performed in all cases.

The online version of this article includes the following figure supplement(s) for figure 7:

**Figure supplement 1.** Maternal effect on the size of the developing olfactory epithelium.

derivatives, in the reciprocal hybrids and their parental morphotypes (*Figure 8C and D*, respectively). In all four cases, the analyses showed strong significant differences between cavefish and surface fish, as previously described (*Alié et al., 2018*) (*Figure 8A–D* histograms).

For *Lhx9* and *Lhx7*, hybrids values were similar and intermediate between those of the cave and surface morphs, with a slight deviation towards the surface morph more evident for the HybSF (*Figure 8A and B*, center and right). For the Hypocretin and NPY neuropeptidergic lineages that are derived from *Lhx9*- and *Lhx7*-expressing progenitors, respectively, a significant difference in neuron numbers existed between reciprocal hybrids (*Figure 8C and D*), suggesting an involvement of maternal components. Moreover, the number of Hypocretin neurons in HybSF and the number of NPY neurons in HybCF were identical those in their maternal morphotype, whereas values for their reciprocal hybrids were close to the theoretical intermediate value (*Figure 8C and D*, on the right). These results suggest that maternal determinants impact hypothalamic neuronal differentiation, possibly together with other, complex, allelic dominance or zygotic mechanisms.

Taken together, these results indicate that the effect of maternal determinants are fully penetrant up until the final stages of gastrulation, suggesting that RNAs and proteins that are present in the oocyte must vary between the two *Astyanax* morphotypes. At later stages, the maternal effect

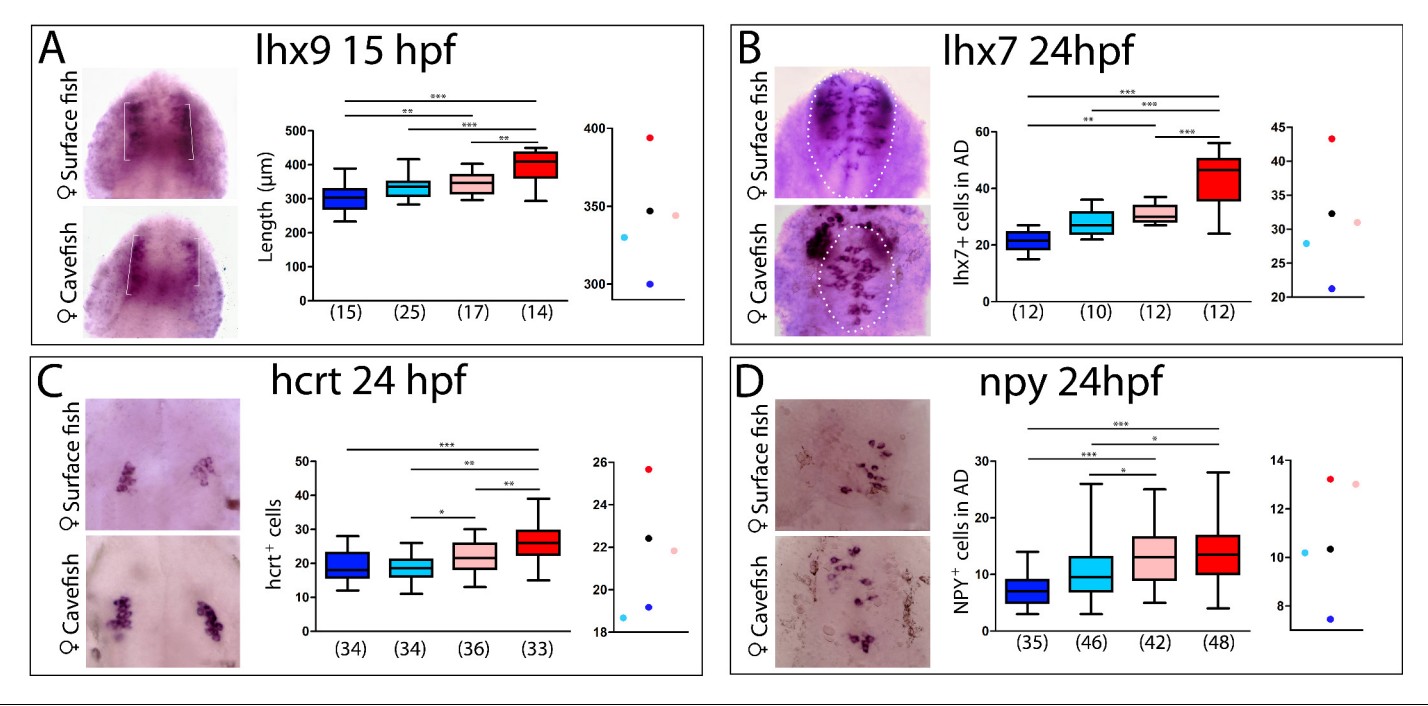

**Figure 8.** Maternal effect on the development of peptidergic systems. (**A**) Expression of *lhx9* in HybSF (left, on top) and HybCF (left, on bottom) at 15 hpf. Quantification of the length of the expression domain in the prospective hypothalamus (white brackets) (center) and the corresponding plot of means distribution (right). (**B**) Expression of *lhx7* in HybSF (left, on top) and HybCF (left, on bottom) at 24 hpf, with the acroterminal domain indicated in dashed lines. Quantification of the number of *lhx7*-expressing cells in the acroterminal domain (center) and the corresponding plot of means distribution (right). (**C**) Expression of *hcrt* HybSF (left, on top) and HybCF (left, on bottom) at 24 hpf. Quantification of the number of hypothalamic *hcrt*-expressing cells (center) and the corresponding plot of means distribution (right). (**D**) Expression of *NPY* in HybSF (left, on top) and HybCF (left, on bottom) at 24 hpf in the acroterminal domain. Quantification of the number of *NPY*-expressing cells (center) and the corresponding plot of means distribution (right). Kruskal-Wallis tests with Dunn's post-test were performed in all cases.

appears to be 'diluted' by other mechanisms that regulate gene expression and morphogenesis, but some important differences can still be observed between reciprocal F₁ hybrids, highlighting how maternal influence translates into later morphological phenotypes.

## Towards the identification of varying maternal factors in cavefish

To obtain an exhaustive molecular view of maternal transcriptomic differences between surface and cavefish, RNA-sequencing was performed on *Astyanax* embryos at the 2-cell stage (surface fish, n = 2 samples; cavefish, n = 3 samples; and reciprocal F₁ hybrids, n = 3 samples each). The dataset (between 75 and 100 million paired reads per sample) was analyzed through the European Galaxy Server and reads were aligned to the Surface Fish *Astyanax* genome (NCBI, GCA_000372685.2 Astyanax_mexicanus-2.0). The sample-to-sample distance analysis grouped the four types of samples into two clear categories, strictly depending on their maternal contribution (***Figure 9A***). Similarly, principal component analyses (PCA) analyses clustered the samples from hybrid embryos together with those from their maternal morphotype (***Figure 9—figure supplement 1A***). These results clearly confirmed that the paternal contribution has no influence on the egg transcriptome at this stage, so we decided to combine the samples according to their mother morphotype (pooled surface fish and hybSF *versus* pooled cavefish and hybCF), thus increasing the number of samples per condition, and rendering downstream analysis easier and more powerful. To compare the transcriptomes of cave and surface eggs quantitatively, the numbers of DEGs were assessed (see Materials and methods). Among the 20,730 genes that were expressed at the 2-cell stage, close to a third (32%) were differentially expressed between surface and cavefish (***Figure 9B***). A similar proportion was up- or down-regulated in cavefish relative to surface fish (17.25% and 14.69%, respectively). To gain insights into which biological functions differed the most between eggs of the two morphotypes, a gene

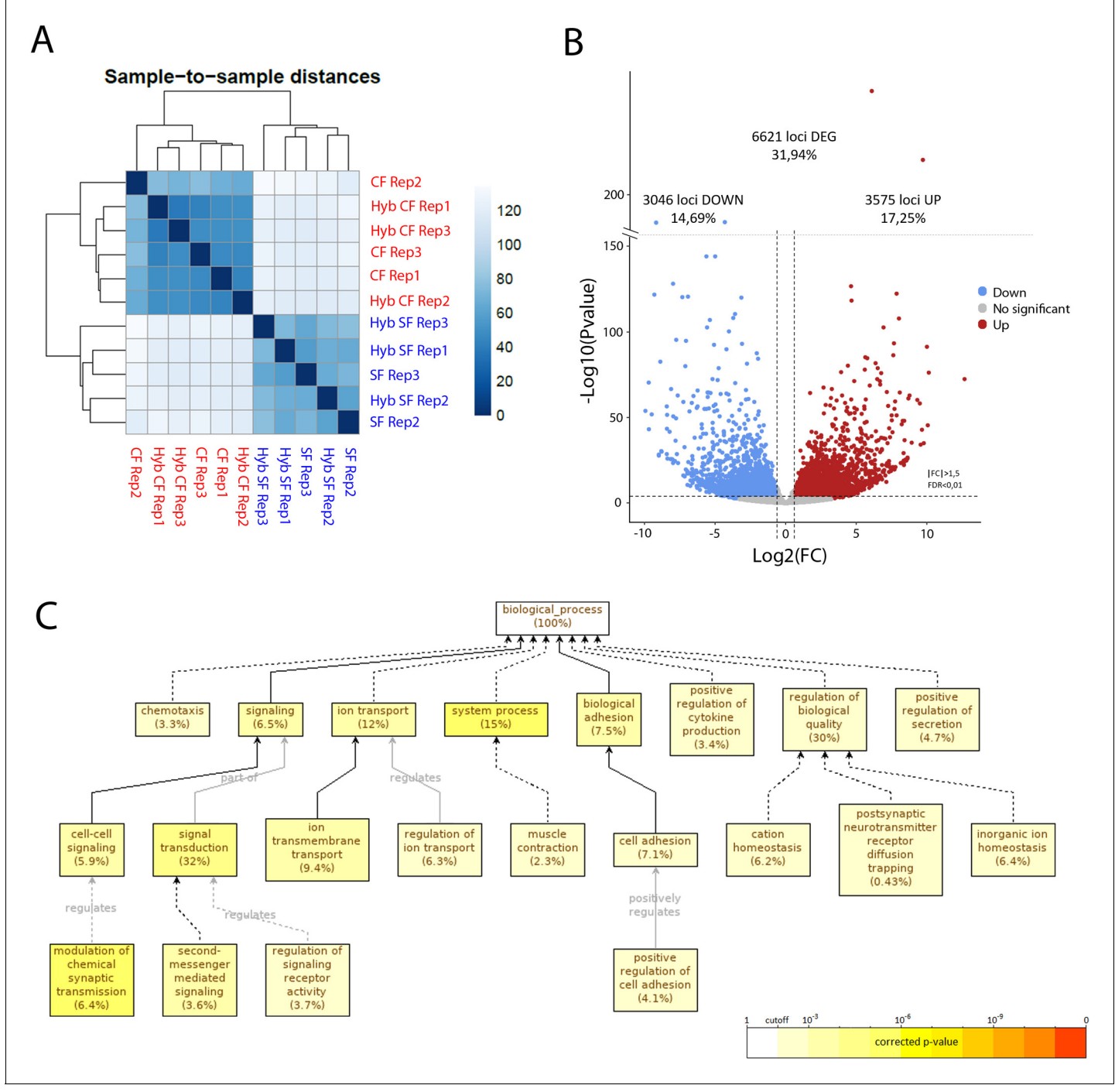

**Figure 9.** RNA-sequencing of the maternal mRNA of surface fish (SF), cavefish (CF) and reciprocal $F_1$ hybrid (HybSF and HybCF) eggs at the 2-cell stage. (A) Sample-to-sample distance between all samples. Samples that are similar are close to each other. On the scale, lower numbers (dark blue) indicate a closer relationship between samples than higher numbers (light blue/white). (B) Volcano plot of expressed genes at the 2-cell stage (n = 20.730). Genes that have an absolute fold change >1.5 and an adjusted p-value (FDR) <0.01 are considered to be differentially expressed in cavefish compared to surface fish. Genes that are upregulated in cavefish are in red, whereas those that are downregulated in cavefish are in blue. (C) Gene ontology enrichment (level: Biological Process) for cavefish DEGs with an absolute fold change higher than 5. Black lines correspond to 'is a' relationship, whereas gray lines correspond to the annotated relationship. Full lines correspond to a direct relationship and dashed lines to an indirect relationship (i.e. some nodes are hidden). The color of a node refers to the adjusted p-value (FDR) of the enriched GO term and the percentage corresponds to the frequency of the GO term in the studied gene set at the level considered. A given gene can have several GO terms. Only enriched GO terms that pass the threshold (p-value<0.01) are displayed on the graph.

The online version of this article includes the following figure supplement(s) for figure 9:

*Figure 9 continued on next page*

*Figure 9 continued*

**Figure supplement 1.** PCA analyses of samples used for RNAseq and GO analyses.

ontology (GO) enrichment analysis was carried out on DEGs with an absolute fold change higher than 5 (log(FC)>2.32193). Cell adhesion (7.1%) and signaling (6.5%) were among the significantly enriched biological processes that might be most relevant for this work (*Figure 9C*). When analyzing up- and downregulated genes for GO enrichment separately, no biological process was found to be enriched in downregulated genes, whereas the above-mentioned processes were still found to be enriched in the upregulated gene subset (*Figure 9—figure supplement 1B*. This means that genes that are involved in ion transport, cell adhesion and cell signaling are mainly upregulated in cavefish eggs compared to surface fish eggs. It is also worth noting that genes that are involved in metabolism show significant enrichment when analyzing all of the DEGs (fold change higher than 1.5), meaning that 'metabolic' transcripts mostly show fold changes lower than 5 (not shown). Hence, the most strongly dysregulated genes are not the ones involved in metabolism but those involved in signaling and cell interactions. Together, these results show that the RNA composition of the cavefish and surface fish eggs shows a strong maternal signature, and thus that oocyte content could contribute to the developmental evolution of cavefish phenotype.

Finally, we picked two candidate genes from the transcriptomics dataset that were directly relevant to our findings in the previous section: (i) *Oep* (*one-eyed pinhead,* also named *tdgf1*), a Nodal co-receptor necessary for *dkk1b* induction and shield formation, whose maternal and zygotic mutant (*MZoep*) shows defects in margin internalization and fate specification in zebrafish (*Carmany-Rampey and Schier, 2001*; *Zhang et al., 1998b*); and (ii) the maternal ventralizing transcription factor *Vsx1* (*Visual System homeobox 1*), which regulates *flh* and *ntl* expression and is involved in axial *versus* paraxial mesoderm specification and migration (*He et al., 2014*; *Xu et al., 2014*). qPCR analyses on 2 hpf embryos showed that *Vsx1* and *Oep* mRNA levels were significantly reduced in cavefish (2.50 and 1.75 times lower expression in cavefish, respectively) confirming the RNA-seq results (8.21 and 1.63 times lower expression, respectively). To test for a possible role of these two downregulated maternal transcripts in the cavefish gastrulation phenotype, we performed overexpression experiments through mRNA injection at the 1-cell stage in cavefish eggs. As read-out of these rescue experiments, *dkk1b* expression was examined at 50% and 70% epiboly. *Vsx1*-injected and *Oep*-injected embryos were similar to control cavefish embryos in terms of their spatio-temporal *dkk1b* pattern, although some signs of disorganization were visible on several specimens (not shown). Thus, a role for *Vsx1* and *Oep* maternal transcripts in the variations of *dkk1b* expression observed between the two *Astyanax* morphs is unlikely. Future experiments should focus on transcripts showing high fold-changes of expression between cavefish and surface fish.

## Discussion

*Astyanax mexicanus* has become an excellent model in which to uncover the developmental mechanisms leading to phenotypic evolution. Modifications in midline signaling centers during early embryogenesis have led to troglomorphic adaptations in cavefish, including eye degeneration, larger olfactory epithelia and an increased number of taste buds. Here, we show striking temporal, spatial and quantitative differences in the expression of the Wnt inhibitor *dkk1b* at the shield stage and during gastrulation, and we explore the idea that maternally regulated gastrulation might be a source of variation that has contributed to cavefish morphological evolution.

### Prechordal plate and forebrain patterning

Genetic manipulations, tissue ablation and transplantation experiments have demonstrated the importance of the prechordal plate as a signaling center involved in the patterning of the basal forebrain (*Heisenberg and Nüsslein-Volhard, 1997*; *Pera and Kessel, 1997*). In fish, the prechordal plate is organized into two domains: the rostral polster (*Kimmel et al., 1995*) and a posterior domain, abutting caudally with the notochord. In *A. mexicanus*, the expression of *shh* in the posterior prechordal plate occupies a wider domain in the cavefish than in the surface fish (*Pottin et al., 2011*; *Yamamoto et al., 2004*), and enhanced shh signaling has pleiotropic effects in the

development of head structures in the cavefish (*Yamamoto et al., 2009*). Here, we showed that the anterior domain of the prechordal plate is a source of the morphogen dkk1b, whose expression is complementary to that of *shh* at the neural plate stage (*Figure 4A–B*). At this stage, *dkk1b*-expressing cells are organized as a compact cluster in surface fish, whereas in cavefish they are more loosely distributed, and with lower levels of *dkk1b* transcripts. Inhibition of Wnt signaling in the presumptive anterior brain is critical for patterning and morphogenesis. Mouse or *Xenopus* embryos with impaired Dkk1 function lack anterior brain structures (*Glinka et al., 1998*; *Mukhopadhyay et al., 2001*), whereas misexpression of *dkk1b* in zebrafish embryos produces anteriorization of the neurectoderm, including enlargement of eyes (*Shinya et al., 2000*). In *Astyanax* also, we found that Wnt activation in surface fish embryos by LiCl-treatments, phenocopying the naturally occurring cavefish condition in which lower levels of *Dkk1b* transcripts could lead to lower Wnt inhibition, leads to a reduction of eye and lens size (*Figure 7C–J*) and an expansion of optic stalk tissue (*Figure 7K–O*), both cavefish-specific hallmarks of eye development (*Yamamoto et al., 2004*; *Devos et al., 2019*). Head development is sensitive to Wnt signaling dosage (*Lewis et al., 2008*), and the temporal variations of *dkk1b* expression that we observed here might contribute to forebrain evolution in cavefish. Indeed, the timing and intensity of Wnt (this work) and Bmp (*Hinaux et al., 2016*) signaling at the anterior pole of the axial mesoderm must instruct the fate and morphogenetic movements of overlying anterior neural plate progenitors that are destined to form the optic region and the hypothalamus, as well as the placode derivatives (*Bielen et al., 2017*; *Rétaux et al., 2013*).

## Embryonic axis formation

The establishment of the embryonic axes and primordial germ layers occurs through complex morphogenetic cell rearrangements during gastrulation (*Schier and Talbot, 2005*; *Solnica-Krezel and Sepich, 2012*). The main outcomes of gastrulation are the spreading of the blastodermal cells, the internalization of endomesoderm precursors and the elongation of the antero-posterior embryonic axis. We hypothesized that the differences observed in the axial mesoderm of *A. mexicanus* morphotypes may be the consequence of upstream events during gastrulation. At equivalent stages, as judged by the percentage of epiboly, we observed that the advancement of internalized tissues migrating in the vegetal to animal direction is more precocious in cavefish embryos than in surface fish embryos. Interestingly, this finding was not only restricted to axial mesodermal elements but also applied to laterally adjacent paraxial mesoderm (*Figure 3*), suggesting a global phenomenon. From the different measurements performed, we inferred that dorsal convergence and anteroposterior extension might be the driving forces that lead to the more advanced phenotype observed in cavefish gastrulas. Interestingly, the differences in hypoblast movements (relative to the percentage of epiboly) that we observed highlight the uncoupling of gastrulation cell movements and the epiboly itself, as spectacularly illustrated in the extreme example of annual killifish embryogenesis (*Pereiro et al., 2017*). We suggest that these temporal variations in gastrulation events might later correlate to differences observed in the off-set of *dkk1b* expression, starting in cavefish before the 6-somite stage and in surface fish after the 8-somite stage.

## Cellular interactions during gastrulation

Gastrulation involves dynamic interactions between different cell populations, and as they move, cells are exposed to changing signals in their immediate environment. Individual interactions between tissues, such as the migration of the hypoblast using epiblast as substrate (*Smutny et al., 2017*) and the influences that the blastodermal cells receive from direct physical contact with the extraembryonic enveloping layer (EVL) (*Reig et al., 2017*) and yolk syncytial layer (YSL) (*Carvalho and Heisenberg, 2010*), must be integrated as gastrulation proceeds. In addition, the prechordal plate has been described as a cell population undergoing collective migration, implying numerous cell–cell interactions between prechordal cells themselves (*Dumortier et al., 2012*; *Zhang et al., 2014*). Genetic dissection of the parameters that regulate prechordal plate migration (*Kai et al., 2008*), as well as the identification of intrinsic properties of the moving group (*Dumortier et al., 2012*), have helped in understanding the molecular and cellular aspects that regulate their migration. The markers that we used here to label the prechordal plate during gastrulation suggest that within this domain, *lefty1*-expressing cells follow collective migration as a cohesive group, whereas *dkk1b*+ cells constitute a more dispersed group, especially in the

cavefish (as also recently observed by *Ren et al., 2018*). Moreover, increased Nodal signaling and changed cell distribution have been reported in the organizer in cavefish embryos (*Ren et al., 2018*). Together with our observation of earlier movements of axial mesoderm cells in cavefish, these data suggest that the structural variations in the cavefish prechordal plate may relate to differential physical and adhesion properties of the organizer/prechordal cells in the two morphs. Live imaging will be necessary to compare better the properties of prechordal plate cells in cavefish and surface fish. Moreover, detailed analyses of the expression of molecules involved in cell adhesion, such as snails and cadherins (*Blanco et al., 2007*; *Montero et al., 2005*; *Shimizu et al., 2005*), as well as those involved in membrane protrusion formation, such as β-actin (*Giger and David, 2017*), will help to explore the possibility that divergence in the intrinsic properties of prechordal plate cells may account for cavefish phenotypic evolution.

## Maternal control of gastrulation and morphological phenotypes

Regardless of the striking morphological evolution observed in *A. mexicanus* morphotypes, their time of divergence has been estimated to be recent (less than 20,000 years ago) (*Fumey et al., 2018*). The inter-fertility of the two morphs, which reflects the short divergence time between them, has allowed the use of hybrids for the identification of the genetic basis behind phenotypic change (*Casane and Rétaux, 2016*; *Protas et al., 2006*).

As early embryonic development is driven by maternal determinants that are present in the oocyte before fecundation, the cross fertility of *A. mexicanus* species is a valuable tool that can be used to obtain information about the contribution of maternal effect genes to phenotypic evolution (*Ma et al., 2018*). Our analyses in $F_1$ reciprocal hybrids demonstrate that the modifications in cavefish gastrulation are fully dependent on maternal factors. In line with this, RNAseq analyses showed that the RNA composition of cavefish and surface fish eggs varied greatly, with 31.94% of the maternal genes that are expressed at the 2-cell stage having differences in transcripts levels. Together, these data strongly suggest that egg composition is a source of variation that can contribute to phenotypic evolution. In both RNAseq and qPCR analyses, the candidate genes beta-catenin 1 and 2, which are involved in the establishment of the organizer (*Kelly et al., 2000*), did not show significantly different levels of expression (not shown). By contrast, two other genes, *oep* and *vsx1*, which are implicated in the development of the prechordal plate (*Gritsman et al., 1999*; *Xu et al., 2014*), showed reduced transcript levels in cavefish compared to surface fish. However, overexpression of these two candidate genes by mRNA injection in cavefish was not able to recapitulate the gastrulation phenotype observed in the surface fish (not shown). *Ma et al. (2018)* have also recently described increased *pou2f1b*, *runx2b*, and *axin1* mRNA levels in unfertilized cavefish eggs as compared to surface fish eggs. These genes also show differential expression in our transcriptomic dataset. Classification of DEGs on the basis of their biological role showed an enrichment in certain biological processes that may have been key for cavefish evolution. Relevant to this work, we found that 6.5% of the 'top DEGs with fold-change >5' are involved in signaling (*Figure 9C*). Some of these genes are regulators of the Wnt pathway (i.e. sFRP2, dkk2, and wnt11) that are important for the establishment of the embryonic organizer. Members of other signaling pathways are also greatly modified (i.e. FGF, BMP, and Nodal). Our interpretation is that the origin of the induction of organizers with different properties in the two morphs might stem from an upstream maternally regulated event, with a domino effect leading to morphological and functionally diverse brains.

Our results on the impact of maternal determinants in later eye and forebrain morphogenesis are puzzling: the retina and the hypothalamus ,which are neural plate derivatives, are under maternal influence for their 'patterning', but the lens and the olfactory epithelium, which are placodal derivatives, are not.

In agreement with our previous independent findings at 36 hpf (*Hinaux et al., 2017*), but contrary to the finding of *Ma et al. (2018)*, here we found that both lens size and lens apoptosis are not under maternal genetic control when assessed at 24, 48, or 72 hpf. A possible explanation for this discrepancy lies in the different markers used to assess lens cell death. We have chosen immunofluorescence staining with an apoptosis-specific marker (activated Caspase3), which allows unambiguous and easy quantification, whereas *Ma et al. (2018)* used a vital staining (LysoTracker) that labels apoptosis, necrosis and autophagy (*Alunni et al., 2007*). Our lens morphometric and Caspase3 data fit well with the absence of maternal effect on the olfactory placode, the placode adjacent to the lens placode which inversely responds to Shh signaling (*Hinaux et al., 2016*), and with the absence of

maternal influence on *Shh* expression in the axial mesoderm, which is thought to control lens apoptosis indirectly (*Yamamoto et al., 2004*). From these results, we conclude that there is little if any contribution of the maternal determinants on the lens defect and degeneration process in the cavefish eye. Clearly, studies are needed in order to dissect the exact molecular mechanism that triggers lens cell death and retinal degeneration in cavefish. This may shade some light on the differential findings reported in *Ma et al. (2018)* and the present study.

Conversely, we show evidence of significant maternal influence on the developmental evolution of the anterior neural plate derivatives in cavefish. At this level, maternal and gastrulation differences translate into important phenotypic outcomes at later larval stages. First, retinal morphogenesis control, including the typical cavefish coloboma assessed by the expression of *Pax2a*, appears to be maternally controlled. We propose that Dkk1b/Wnt signaling, itself fully maternally controlled from the beginning of gastrulation onwards (*Figures 1*, *2* and *4*) and whose manipulation affects the ventral eye phenotype (*Figure 5*), is involved in the process. Of note, this result is in line with those of *Ma et al. (2018)*, who reported a qualitative maternal genetic effect on the ventral position of the lens. Second, the evolution of hypothalamic neuronal patterning, specifically the patterning of Hypocretin and NPY neurons which begin differentiating around 20 hpf (i.e. long after zygotic genome activation) (*Alié et al., 2018*), also reflects a significant and long-lasting maternal influence.

As regards forebrain development more generally, our interpretation is that although maternal factors greatly influence early developmental decisions, later phenotypes become 'diluted' as other mechanisms enter into play. We suggest that as the zygotic genome takes control of development, allelic dominance has an increasing impact on the phenotypes after 15 hpf onwards, although we did observe some maternally controlled phenotypes in hybrids for some relevant traits: it is remarkable that retinal morphology at 2 dpf – the coloboma phenotype – is largely dependent on regulations that occurred in the mother's gonad. The same holds true for hypothalamic neuronal patterning, which has very important behavioral and adaptive consequences (*Alié et al., 2018*; *Jaggard et al., 2018*). Of note in *Astyanax*, some behavioral traits in adults have already been shown to be under parental inheritance (*Yoshizawa et al., 2012*): the vibration attraction behavior and its underlying sensory receptors (the neuromasts) are under paternal inheritance in cavefish originating from the Pachón cave, whereas they are under maternal inheritance in cavefish originating from the Los Sabinos cave. These examples underscore the different levels of developmental regulation that must interact to produce a hybrid phenotype.

The study of the impact of maternal components in the morphological and developmental evolution of species is ongoing. To our knowledge, besides *Astyanax* cavefish, only one study has reported a maternal contribution regulating the developmental trajectory of entry into diapause in a killifish (*Romney and Podrabsky, 2017*). Thus *Astyanax* cavefish appear to be a proper model in disentangling the very early genetic and embryonic mechanisms of morphological evolution. In addition, the modified expression of maternal genes could result from differential *cis*-regulation, which, to our knowledge, has not yet been explored for maternal effect genes in the evolutionary context in any species.

## Materials and methods

### A. *mexicanus* embryos

Our surface fish colony originates from rivers in Texas, United States, and our cavefish colony derives from the Pachón cave in San Luis Potosi, Mexico. Embryos were obtained by in vitro fertilization and/or natural spawnings induced by changes in water temperature (*Elipot et al., 2014b*). The development of *A. mexicanus* at 24°C is similar and synchronous for both morphotypes (*Hinaux et al., 2011*). For this study, morphological aspects were taken as strict criteria to stage the embryos (number of cells, percentage of epiboly and number of somites). In vitro fertilizations were performed to generate reciprocal $F_1$ hybrids by fecundating cavefish oocyte with surface fish sperm (HybCF) and surface fish oocyte with cavefish sperm (HybSF).

### Whole-mount in situ hybridization (ISH)

ISH was carried out as previously described (*Pottin et al., 2011*). Digoxigenin- and fluorescein-labeled riboprobes were prepared using PCR products as templates. Genes of interest were

searched in an EST (expressed sequence tag) library that was accessible in the laboratory. Clones in library (pCMV-SPORT6 vector) were: *chordin* (ARA0AAA23YC10), *dkk1b* (ARA0AAA18YA07EM1), *eya2* (ARA0AAA19YL19EM1), *floating-head* (ARA0ACA35YA23), *myoD* (ARA0AAA95YG16), *msgn1* (ARA0ACA49YF15), *no-tail* (ARA0ABA99YL22), *npy* (FO263072), *vsx1* (ARA0AHA13YJ18) and *pax2a* (*Devos et al., 2019*). Other cDNAs that were previously cloned were: *fgf8* (DQ822511), *lhx9* (EF175738), *shh* (AY661431), *dlx3b* (AY661432), *hcrt* (XM_007287820.3), and *lhx7* (XM_022678613). Total RNA from *Astyanax* embryos of various stages (2–24 hpf) was reverse-transcribed using the iScript cDNA synthesis kit (Bio-Rad) and amplified using the following primers: brachyury – forward primer (FP) CACCGGTGGAAGTACGTGAA, reverse primer (RP) GGAGCCGTCGTATGGAGAAG; *lefty1* – FP ACCATGGCCTCGTGCCTC; RP TCAGACCACCGAAATGTTGTCCAC.

Full-length cDNAs were cloned into the pCS2+ expression vector using the indicated restriction sites: *dkk1b* (sites EcoRI and XhoI) — FP GGTGGTGAATTCACCATGTGGCCGGCGGCGCTC TCAGCCCTGACCTTC, RP ACCACCCTCGAGTCAGTGTCTCTGGCAGGTATGG; *vsx1* (sites XhoI and XbaI) — FP GGTGGTCTCGAGACCATGGAGAAGACACGCGCG, RP ACCACCTCTAGATCAG TTCTCGTTCTCTGAATCGC; *oep* (*tdgf1*) (sites BamHI and XbaI) — FP GGTGGTGGATCCACCA TGAGGAGCTCAGTGTTCAGG, RP ACCACCTCTAGATCAAAGCAGAAATGAAAGGAGGAG.

## Immunohistochemistry

Whole-mount immunohistochemistry was performed as previously described in *Blin et al. (2018)*. Rabbit anti-activated Caspase 3 (Abcam, Ab13847-25) was used and goat anti-Rabbit IgG (H+L) coupled with Alexa Fluor 488 (Thermo Fisher Scientific, A-11034) was used as secondary antibody. Samples were counterstained using DAPI (Sigma, 10236276001) and DiI (Thermo Fisher Scientific, D-282), whole mounted in Vectashield medium (Vector, H1000) and imaged with a confocal microscope (Leica TCS SP8). Cell counting and volumetric analyses were performed in Fiji ImageJ.

## mRNA injections

In vitro transcription was carried out from PCR products using the SP6 RNA polymerase (mMESSAGE mMACHINE) to generate full-length capped mRNA. Dilutions of the mRNA to 150–200 ng/μL were prepared in phenol red 0.05%. Embryos at the 1-cell stage were injected with 5–10 nL of working solutions using borosilicate glass pipettes (GC100F15, Harvard Apparatus LTD) pulled in a Narishige PN-30 Puller (Japan).

## LiCl treatments

Embryos were enzymatically dechorionated in 1 mg/mL pronase solution dissolved in electron microscopy (EM) water, then they were incubated in LiCl solutions, 0.1M or 0.2M prepared in EM water, during the indicated time window. After the treatment, embryos were washed five times in EM, and allowed to develop until the stage desired for further analyses.

## Image acquisition and analyses

Whole-mounted embryos stained by colorimetric and fluorescent ISH were imaged on a Nikon AZ100 multizoom macroscope coupled to a Nikon digital sight DS-Ri1 camera, using the NIS software. Mounted specimens were imaged on a Nikon Eclipse E800 microscope equipped with a Nikon DXM 1200 camera running under Nikon ACT-1 software. Confocal images were captured on a Leica SP8 microscope with the Leica Application Suite software. Morphometric analyses and cell counting were performed on the Fiji software (Image J). To measure the approximate extent of migration in the vegetal to animal axis (Height), we measured the distance from the margin to the leading cell normalized by the distance from the margin to the animal end (*Figure 2—figure supplement 1*). To estimate the extent of dorsal convergence, we measured either the width of the expression domain or the width of gap without expression (for example for *myoD*) normalized by the total width of the embryo (a representation using the expression of *myoD* at 70% epiboly is shown in *Figure 3—figure supplement 1*). All measurements were normalized, unless otherwise indicated. Another means that we used to calculate the width of expression was by measuring the angle (α) of the expression pattern from an animal view, using the center of the opposite site to the expression domain to set the vertex (a representation using the expression of *chordin* at 50% epiboly is shown in *Figure 1— figure supplement 1*). To assess the width of the *pax2a* expression domain in the optic stalk/optic

fissure, we measured the angle (α) with the vertex set in the center of the lens (*Figure 5—figure supplement 1*). Statistical analyses were done in Graph pad prism 5. When needed, double blind measurements were performed on anonymized images. All raw quantifications are available in the Supplementary file.

## mRNA isolation

RNA pools were isolated from cavefish, surface fish and $F_1$ hybrid embryos at the 2-cell stage (three independent biological replicates for each condition). Each sample corresponded to at least 20 embryos coming from two female individuals (40 embryos in total) to reduce inter-individual variability. Total RNA was extracted using TRIzol (Invitrogen, 2 µL per embryo) and chloroform (0.2 µL per µL of TRIzol), purified with isopropanol (0.5 µL per µL of TRIzol) and 70% ethanol, and treated with DNase. Following purification, all samples were immediately quantified and assessed for RNA quality (A260/280 ratio ~1.9–2.1) using a NanoVue Spectrophotometer and stored at −80°C until use.

## qPCR

1 µg of total RNA was reverse transcribed in a 20 µL final reaction volume using the High Capacity cDNA Reverse Transcription Kit (Life Technologies) with RNase inhibitor and random primers following the manufacturer's instructions. Quantitative PCR was performed on a QuantStudioTM 12K Flex Real-Time PCR System with a SYBR green detection protocol at the qPCR platform of the Gif CNRS campus. 3 µg of cDNA were mixed with Fast SYBRV R Green Master Mix and 500 nM of each primer in a final volume of 10 µL. The reaction mixture was submitted to 40 cycles of PCR [95°C/20 s; (95°C/1 s; 60°C/20 s) X40] followed by a fusion cycle to analyze the melting curve of the PCR products. Negative controls without the reverse transcriptase were introduced to verify the absence of genomic DNA contaminants. Primers were designed using the Primer-Blast tool from NCBI and the Primer Express 3.0 software (Life Technologies). Primers were defined either in one exon and one exon–exon junction or in two exons span by a large intron. Specificity and the absence of multilocus matching at the primer site were verified by BLAST analysis. The amplification efficiencies of primers were generated using the slopes of standard curves obtained by a four-fold dilution series. Amplification specificity for each real-time PCR reaction was confirmed by analysis of the dissociation curves. Determined Ct values were then exploited for further analysis, with the *Gapdh* gene as reference.

## RNAseq analyses of maternal mRNAs

RNA sequencing was carried out at the I2BC High-throughput sequencing platform (https://www.i2bc.paris-saclay.fr/spip.php?article399) using an Illumina NextSeq 500 sequencing instrument (version NS500446). All RNA samples were checked with a Bioanalyzer RNA 6000 pico chip (Agilent technologies) and passed the quality threshold (RIN >9) prior to library preparation. Libraries were generated from purified total RNA using polyA selection (Ilumina TruSeq Stranded Protocol). Samples were sequenced for between 75 and 100 million reads (paired-end, 51–35 bp) using the NextSeq 500/550 High Output Kit v2 (75 cycles). Following sequencing, raw data were retrieved (fastq-formatted files) and used for subsequent sequence alignment and expression analyses. Raw sequencing data are available through the NCBI Sequence Reads Archive (SRA) under BioProject accession PRJNA545230.

RNA-sequencing reads from each of the four conditions (surface fish, cavefish and reciprocal $F_1$ hybrids) were trimmed using Cutadapt 1.15 and quality control was assessed using FastQC (v0.11.5). All downstream analyses were done using the European Galaxy Server (https://usegalaxy.eu; *Afgan et al., 2016*) with reverse (RF) strandness parameter. Reads were aligned to the Surface Fish *Astyanax* genome (NCBI, GCA_000372685.2 Astyanax_mexicanus-2.0) using HISAT2 (Galaxy Version 2.1.0; *Kim et al., 2015*) and only perfectly aligned paired reads were kept for the following analysis (Filter SAM and Bam file Galaxy Version 1.8: Minimum MAPQ quality score 20 and Filter on bitwise flag 'Read is paired' and 'Read is mapped in a proper pair'). Then, aligned reads were counted using htseq-count (Galaxy Version 0.9.1; *Anders et al., 2015*) and the *A. mexicanus* annotation from NCBI release 102 (https://www.ncbi.nlm.nih.gov/genome/?term=txid7994[orgn]). Genes that were differentially expressed (DEG) between cavefish and surface fish were detected using DESeq2 (Galaxy Version 2.11.40.6; *Love et al., 2014*). On the basis of the PCA and sample-to-sample distance

analyses, data from $F_1$ hybrids were combined with their respective mother morphotype for the DEG analysis. Only genes with a FDR <0.01 (p-value adjusted for multiple testing with the Benjamini-Hochberg procedure) and absolute fold change higher than 1.5 ($\log_2$(FC) >0.58) were kept as significantly over- or underexpressed in cavefish compared to surface fish. Mapped reads were visualized using the genome browser IGV (http://www.broadinstitute.org/igv/) (*Robinson et al., 2011*).

A Gene Ontology Annotation file for the *Astyanax* transcriptome (from the NCBI database) was generated using OmicsBox (formerly Blast2GO, https://www.biobam.com/) following the general workflow presented by the software: BLAST with CloudBlast [restricted to the teleosteii database, keeping the top 20 results with an e-value of 10(−5)], followed by mapping (GO version April 2019), annotation and InterProScan analysis in parallel. The annotation file was generated by merging the annotated BLAST results with InterProScan results. Gene Ontology Enrichment analysis on DEG (FDR <0.01 and FC >1.5) was carried out on Galaxy using GOEnrichment (Galaxy Version 2.0.1) with a p-value cut-off of 0.01. We used several thresholds of fold change (FC >1.5, FC >5, FC >10, FC >20 and FC >50) to define gene sets and performed the analysis using the genes that were expressed at the 2-cell stage as reference (n = 20,730). In this study, the study gene set with FC >5 was kept as it is the most biologically meaningful.

## Acknowledgements

We thank Stéphane Père, Victor Simon and Krystel Saroul for care of our *Astyanax* colony, and François Agnès and all other members of the DECA team for fruitful discussions and important suggestions on our study. This work has benefited from the facilities and expertise of the QPCR and the sequencing platforms of I2BC, Gif sur Yvette. We thank Yan Jaszczyszyn from the I2BC sequencing platform for fruitful interactions. Grant support was received from FRM (Equipe FRM DEQ20150331745 RETAUX), CNRS and Becas Chile.

## Additional information

### Funding

| Funder | Grant reference number | Author |
|---|---|---|
| Agence Nationale de la Recherche | blindtest | Sylvie Rétaux |
| Fondation pour la Recherche Médicale | DEQ20150331745 RETAUX | Sylvie Rétaux |

The funders had no role in study design, data collection and interpretation, or the decision to submit the work for publication.

### Author contributions

Jorge Torres-Paz, Conceptualization, Formal analysis, Investigation, Visualization, Methodology, Writing—original draft, Writing—review and editing; Julien Leclercq, Conceptualization, Formal analysis, Investigation, Visualization, Methodology, Writing—original draft; Sylvie Rétaux, Conceptualization, Resources, Supervision, Funding acquisition, Validation, Investigation, Visualization, Methodology, Writing—original draft, Writing—review and editing

### Author ORCIDs

Jorge Torres-Paz  https://orcid.org/0000-0002-7277-6348
Sylvie Rétaux  https://orcid.org/0000-0003-0981-1478

### Ethics

Animal experimentation: Animal experimentation: Animals were treated according to the French and European regulations for handling of animals in research. SR's authorization for use of animals in research including Astyanax mexicanus is 91-116 and Paris Centre-Sud Ethic Committee authorization numbers are 2012- 0052, -0053, and -0054.

**Decision letter and Author response**
Decision letter https://doi.org/10.7554/eLife.50160.SA1
Author response https://doi.org/10.7554/eLife.50160.SA2

## Additional files

### Supplementary files

• Supplementary file 1. Supplementary tables including the raw quantifications.
• Transparent reporting form

### Data availability

Raw sequencing data are available through the NCBI Sequence Reads Archive (SRA) under BioProject accession PRJNA545230.

The following dataset was generated:

| Author(s) | Year | Dataset title | Dataset URL | Database and Identifier |
|---|---|---|---|---|
| Torres-Paz J, Leclercq J, Rétaux S | 2019 | Maternal RNA sequencing of Astyanax mexicanus 2-cells eggs | https://www.ncbi.nlm.nih.gov/sra/PRJNA545230 | NCBI Bioproject, PRJNA545230 |

The following previously published datasets were used:

| Author(s) | Year | Dataset title | Dataset URL | Database and Identifier |
|---|---|---|---|---|
| Hinaux H, Poulain J, Da Silva C, Noirot C, Jeffery WR, Casane D, Retaux S | 2012 | FO221961 Astyanax mexicanus whole embryos and larvae neurula to swimming larvae Astyanax mexicanus cDNA clone ARA0AAA23YC10, mRNA sequence | https://www.ncbi.nlm.nih.gov/nuccore/422274646 | NCBI Nucleotide, 422274646 |
| Hinaux H, Poulain J, Da Silva C, Noirot C, Jeffery WR, Casane D, Retaux S | 2012 | FO212120 Astyanax mexicanus whole embryos and larvae neurula to swimming larvae Astyanax mexicanus cDNA clone ARA0AAA18YA07, mRNA sequence | https://www.ncbi.nlm.nih.gov/nuccore/422224244 | NCBI, 422224244 |
| Hinaux H, Poulain J, Da Silva C, Noirot C, Jeffery WR, Casane D, Retaux S | 2012 | FO211529 Astyanax mexicanus whole embryos and larvae neurula to swimming larvae Astyanax mexicanus cDNA clone ARA0AAA19YL19, mRNA sequence | https://www.ncbi.nlm.nih.gov/nuccore/422226109 | NCBI Nucleotide, 422226109 |
| Hinaux H, Poulain J, Da Silva C, Noirot C, Jeffery WR, Casane D, Retaux S | 2012 | FO304658 Astyanax mexicanus whole embryos and larvae neurula to swimming larvae Astyanax mexicanus cDNA clone ARA0ACA35YA23, mRNA sequence | https://www.ncbi.nlm.nih.gov/nuccore/425531810 | NCBI Nucleotide, 425531810 |
| Hinaux H, Poulain J, Da Silva C, Noirot C, Jeffery WR, Casane D, Retaux S | 2012 | FO245358 Astyanax mexicanus whole embryos and larvae neurula to swimming larvae Astyanax mexicanus cDNA clone ARA0AAA95YG16, mRNA sequence | https://www.ncbi.nlm.nih.gov/nuccore/422281476 | NCBI Nucleotide, 422281476 |
| Hinaux H, Poulain J, Da Silva C, Noirot C, Jeffery WR, Casane D, Retaux S | 2012 | FO299407 Astyanax mexicanus whole embryos and larvae neurula to swimming larvae Astyanax mexicanus cDNA clone ARA0ACA49YF15, mRNA sequence | https://www.ncbi.nlm.nih.gov/nuccore/425597731 | NCBI Nucleotide, 425597731 |
| Hinaux H, Poulain J, Da Silva C, Noirot C, Jeffery WR, Casane D, Retaux S | 2012 | FO289143 Astyanax mexicanus whole embryos and larvae neurula to swimming larvae Astyanax mexicanus cDNA clone ARA0ABA99YL22, mRNA sequence | https://www.ncbi.nlm.nih.gov/nuccore/425545401 | NCBI Nucleotide, 425545401 |

| | | | | |
|---|---|---|---|---|
| Hinaux H, Poulain J, Da Silva C, Noirot C, Jeffery WR, Casane D, Retaux S | 2012 | FO378514 Astyanax mexicanus whole embryos and larvae neurula to swimming larvae Astyanax mexicanus cDNA clone ARA0AHA13YJ18, mRNA sequence | https://www.ncbi.nlm.nih.gov/nuccore/425608699 | NCBI Nucleotide, 425608699 |
| Hinaux H, Poulain J, Da Silva C, Noirot C, Jeffery WR, Casane D, Retaux S | 2012 | FO263072 Astyanax mexicanus whole embryos and larvae neurula to swimming larvae Astyanax mexicanus cDNA clone ARA0ABA37YE05, mRNA sequence | https://www.ncbi.nlm.nih.gov/nuccore/FO263072 | NCBI Nucleotide, FO263072 |
| Menuet A, Alunni A, Joly JS, Jeffery WR, Rétaux S | 2007 | Astyanax mexicanus LIM/homeobox protein 9 mRNA, complete cds | https://www.ncbi.nlm.nih.gov/nuccore/EF175738 | NCBI Nucleotide, EF175738.1 |
| Yamamoto Y, Stock DW, Jeffery WR | 2004 | Astyanax mexicanus sonic hedgehog precursor (shh) mRNA, complete cds | https://www.ncbi.nlm.nih.gov/nuccore/AY661431 | NCBI Nucleotide, AY661431 |
| Yamamoto Y, Stock DW, Jeffery WR | 2004 | Astyanax mexicanus distal-less homeobox gene 3b (dlx3b) mRNA, partial cds | https://www.ncbi.nlm.nih.gov/nuccore/AY661432 | NCBI Nucleotide, AY661432 |
| Warren W, McCaugh S | 2017 | PREDICTED: Astyanax mexicanus hypocretin neuropeptide precursor (hcrt), mRNA | https://www.ncbi.nlm.nih.gov/protein/XM_007287820 | NCBI Nucleotide, XM_007287820.3 |
| Warren W, McCaugh S | 2017 | PREDICTED: Astyanax mexicanus LIM homeobox 8 (lhx8), transcript variant X1, mRNA | https://www.ncbi.nlm.nih.gov/nuccore/XM_022678613 | NCBI Nucleotide, XM_022678613.1 |

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
