## [Decision Letter]

**Acceptance summary:**

This paper describes maternally-controlled differences in gastrulation between surface- and cave-dwelling forms of Astyanax. The paper is well written, and the data were carefully collected and clearly presented. The paper provides ample evidence of gastrulation- and axis formation-specific gene expression patterns to support this conclusion. Further, they use reciprocal hybrids of the two forms to illustrate that differences in gene expression during gastrulation are due to maternal influences. This paper represents an important contribution to the literature and strengthens the concept that maternal packaging has the ability to effect alternative developmental outcomes in vertebrates.

**Decision letter after peer review:**

Thank you for submitting your article "Maternally-regulated gastrulation as a source of variation contributing to cavefish forebrain evolution" for consideration by *eLife*. Your article has been reviewed by two peer reviewers, and the evaluation has been overseen by Marianne Bronner as the Senior and Reviewing Editor.

The reviewers have discussed the reviews with one another and the Reviewing Editor has drafted this decision to help you prepare a revised submission.

Summary:

This paper describes maternally-controlled differences in gastrulation between surface- and cave-dwelling forms of Astyanax. The paper provides evidence of gastrulation- and axis formation-specific gene expression patterns to support this conclusion. Further, they use reciprocal hybrids of the two forms to illustrate that differences in gene expression during gastrulation are due to maternal influences. First, they identify members of the WNT signaling to be differentially expressed during early development, arguing for a role of WNT signaling in the eye and forebrain differences between the two morphs. Second, using elegant reciprocal hybrid crosses, they find that these differences are almost entirely due to a maternal contribution (at least in the earliest stages). The maternal contribution extends also to other previously identified differences in hypothalamic neuronal differentiation. However, there was concern that, without a stronger phenotype, the paper may be better suited for a specialized journal.

Essential revisions:

1) The most interesting part of this paper is the fact that maternal factors are having effects long after the zygotic genome is activated and thus are affecting developmental patterns. Unfortunately the maternal contributions do not translate in any appreciable phenotypic differences with the exception of some very small lens phenotypes. Figure 7 shows embryos at 24 hpf, when the phenotypes are extremely similar between cave and surface fish and the eyes degenerate later in development in the cavefish. The authors need to be more clear about this.

2) Importantly, the authors should look at 48 hpf or 72 hpf when to phenotypes are more divergent to see if there is a stronger phenotype. These experiments should be done double blind and with appropriate power and sample size calculations.

---

## [Author Response]

Essential revisions:1) The most interesting part of this paper is the fact that maternal factors are having effects long after the zygotic genome is activated and thus are affecting developmental patterns. Unfortunately the maternal contributions do not translate in any appreciable phenotypic differences with the exception of some very small lens phenotypes. Figure 7 shows embryos at 24 hpf, when the phenotypes are extremely similar between cave and surface fish and the eyes degenerate later in development in the cavefish. The authors need to be more clear about this.

We have now performed series of novel experiments to respond to this important question. The data are presented in the section “Maternal determinants influence late phenotypes in *A. mexicanus* morphotypes” and the new Figure 7. We have investigated the “eye phenotype”, which is the most striking cavefish phenotype, and sorted out effects on the lens and effects on the retina, which developmentally derive from distinct origins (placodal versus neural).

Regarding the lens, we found that two independent aspects of its phenotype, the small size and the apoptotic process, are not different in reciprocal F1 hybrids at 1, 2, or 3 days post-fertilization. This suggests that these phenotypes are not maternally-controlled in cavefish. Of note, our results contrast with recently published data from Ma et al., 2018), who described maternal genetic effects on both the lens size and the lens apoptosis. 1) Regarding the size, our present results are in agreement with previous results from our group obtained by independent investigators on independent samples (Hinaux et al., 2017). Our results are also consistent with the fact that the size of the olfactory placode, which derives from the placodal territory adjacent to the lens and whose sizes are inversely correlated in cavefish and surface fish embryos, is similar as well in reciprocal F1 hybrids. 2) Regarding apoptosis, we are confident in our data obtained with the apoptosis-specific Caspase3 marker which allows unambiguous quantification of apoptotic cells (as opposed to the non-specific and difficult to quantify in our hands, LysoTracker used by Ma et al., 2018; Figure 3M, N, O of their manuscript). In addition, we found the same results at both 2 and 3dpf in the lens with very large samples, we found the same trend in the 3dpf retina, and our comparison with the surface fish and the cavefish phenotypes show evidence for a partial dominance of the surface fish zygotic program (only F1 hybrids are shown in Ma et al., 2018). Finally, and importantly, as lens apoptosis is indirectly controlled by *Shh* overexpression in cavefish (Yamamoto et al., 2004), we checked for *Shh* expression in reciprocal F1 hybrids and found no evidence for maternal control at this level either. We therefore concluded that the size and the apoptosis of the lens in cavefish does not appear to be maternally-regulated.

Regarding the retina, we measured a critical aspect of its phenotype in cavefish: the defective morphogenesis, including a ventral coloboma. Using the *Pax2a* optic stalk/central fissure marker in 48hpf larvae, we detected a very significant maternal effect on this coloboma phenotype. This result is very coherent with the data shown earlier in our manuscript regarding the maternally-regulated gastrulation and expression of *Dkk1b*, and the effects of the manipulation of Wnt signaling on the ventral eye phenotype. Of note, this result is here in line with those of Ma et al., 2018, who reported qualitatively a maternal genetic effect on the ventral position of the lens (Figure 2D, F of their manuscript). Together with the results on neuropeptidergic patterning in the hypothalamus that were already presented in the original version of our manuscript, this new section now shows that maternally regulated gastrulation as long-lasting effects, with clear phenotypes in 48hpf larvae, on the developmental evolution of the cavefish forebrain.

2) Importantly, the authors should look at 48 hpf or 72 hpf when to phenotypes are more divergent to see if there is a stronger phenotype. These experiments should be done double blind and with appropriate power and sample size calculations.

As explained above, we have now looked at late phenotypes at 48 and 72hpf. Indeed, the data have been obtained from at least 2 independent experiments, each including for the 4 different samples (surface fish, cavefish, and the 2 reciprocal F1 hybrids) embryos obtained from different spawnings to avoid mother-specific effects. Sample sizes are very high (ex: n=35-42 for the *Pax2a* phenotype). And very importantly, analyses were performed in double blind (by JTP and SR) on anonymized images.